# Sustainable Pest Management Using Novel Nanoemulsions of Honeysuckle and Patchouli Essential Oils against the West Nile Virus Vector, *Culex pipiens*, under Laboratory and Field Conditions

**DOI:** 10.3390/plants12213682

**Published:** 2023-10-25

**Authors:** Wafaa M. Hikal, Mohamed M. Baz, Mohammed Ali Alshehri, Omar Bahattab, Rowida S. Baeshen, Abdelfattah M. Selim, Latifah Alhwity, Rabaa Bousbih, Maha Suleiman Alshourbaji, Hussein A. H. Said-Al Ahl

**Affiliations:** 1Department of Biology, Faculty of Science, University of Tabuk, Tabuk 71491, Saudi Arabia; ma.alshehri@ut.edu.sa (M.A.A.); obahattab@ut.edu.sa (O.B.); rbaeshen@ut.edu.sa (R.S.B.); lalhawiti@ut.edu.sa (L.A.); mahaaallshorbaji123@gmail.com (M.S.A.); 2Parasitology Laboratory, Water Pollution Research Department, Environment and Climate Change Institute, National Research Centre (NRC), 33 El-Behouth St., Dokki, Giza 12622, Egypt; 3Department of Entomology, Faculty of Science, Benha University, Benha 13518, Egypt; mohamed.albaz@fsc.bu.edu.eg; 4Department of Animal Medicine (Infectious Diseases), College of Veterinary Medicine, Benha University, Toukh 13736, Egypt; abdelfattah.selim@fvtm.bu.edu.eg; 5Department of Physics, Faculty of Science, University of Tabuk, Tabuk 71421, Saudi Arabia; rbousbih@ut.edu.sa; 6Medicinal and Aromatic Plants Research Department, Pharmaceutical and Drug Industries Research Institute, National Research Centre (NRC), 33 El-Behouth St., Dokki, Giza 12622, Egypt; shussein272@yahoo.com

**Keywords:** essential oils, plants bioactive compounds, polyphenols, nanoemulsions, *Culex pipiens*, honeysuckle, patchouli

## Abstract

Essential oils are natural plant products that are very interesting, as they are important sources of biologically active compounds. They comprise eco-friendly alternatives to mosquito vector management, particularly essential oil nanoemulsion. Therefore, the aim of this study is to evaluate the effectiveness of 16 selected essential oils (1500 ppm) in controlling mosquitoes by investigating their larvicidal effects against the larvae and adults of the West Nile virus vector *Culex pipiens* L. (Diptera: Culicidae); the best oils were turned into nanoemulsions and evaluated under laboratory and field conditions. The results show that honeysuckle (*Lonicera caprifolium*) and patchouli (*Pogostemon cablin*) essential oils were more effective in killing larvae than the other oils (100% mortality) at 24 h post-treatment. The nanoemulsions of honeysuckle (LC_50_ = 88.30 ppm) and patchouli (LC_50_ = 93.05 ppm) showed significantly higher larvicidal activity compared with bulk honeysuckle (LC_50_ = 247.72 ppm) and patchouli (LC_50_ = 276.29 ppm) oils. *L. caprifolium* and *P. cablin* (100% mortality), followed by *Narcissus tazetta* (97.78%), *Rosmarinus officinalis* (95.56%), and *Lavandula angustifolia* (95.55%), were highly effective oils in killing female mosquitoes, and their relative efficacy at LT_50_ was 5.5, 5.3, 5.8, 4.1, and 3.2 times greater, respectively, than *Aloe vera*. The results of the field study show that the honeysuckle and patchouli oils and their nanoemulsions reduced densities to 89.4, 86.5, 98.6, and 97.0% at 24 h post-treatment, respectively, with persistence for eight days post-treatment in pools. Nano-honeysuckle (100% mortality) was more effective than honeysuckle oils (98.0%). Our results show that honeysuckle and patchouli oils exhibited promising larvicidal and adulticidal activity of *C. pipiens*.

## 1. Introduction

Mosquitoes are widely distributed in temperate, semi-tropical, tropical, and Arctic regions. Globally, there are 41 genera containing more than 3600 different species that have been recognized [1]. Because of their high adaptation and sensitivity, mosquitoes quickly adapt to new habitats, climatic change, the structure of forests, and residential developments. Additionally, because of these features or conditions, some species can thrive in environments that were not previously suitable for them, such as artificial containers, pools, or new locations, which helps spread mosquito reproduction and therefore increases the possibility of disease and death rates [2,3]. 

Some species of mosquito are dangerous to public health because they carry pathogens that make people and animals sick, like the ones that cause malaria, dengue, Zika, chikungunya, Japanese encephalitis, West Nile fever, and yellow fever in people as a result of the bites of a female mosquito [4]; moreover, their bites can result in red rashes, pain, and itching, and scratching them can result in bacterial infection [5]. Less than 5% of all mosquitoes in the world are dangerous carriers, and the majority are important components of aquatic and terrestrial ecosystems [5,6].

*Culex pipiens* (northern house mosquito) is the vector of the West Nile virus (WNV) that causes encephalitis and meningitis. The disease affects the brain tissue, and the most serious cases can result in permanent neurological damage and be fatal [7]. There is no vaccine to prevent this infection, nor are there drugs to combat the disease in infected persons; thus, vector control is the most prevalent solution available so far for reducing morbidity. The most widely used vector interruption methods are synthetic and based on insecticides [8].

Synthetic pesticides have helped substantially in ridding areas of insects that spread diseases, but they may cause some very serious negative effects on people and the environment, such as the emergence of dangerous chemical residues and groundwater and soil pollution [9]. Moreover, pesticide-resistant insects are becoming more common, which has made it important to look for safer natural alternatives, such as microbial insecticides and botanical pesticides that are effective in controlling many insect populations and that are simultaneously cheap and easily biodegradable, easy to obtain, and do not hurt organisms that are not their targets [10].

Botanical pesticides (essential oils, flavonoids, alkaloids, glycosides, esters, and fatty acids) are natural chemicals and a great alternative to chemical or synthetic pesticides, and they have different chemical properties, modes of action, and effects on insects; they are insect repellents, feed inhibitors or antifeedants, toxic substances, growth inhibitors, chemical enhancers, and attractants [11,12]. Therefore, botanical pesticides are used instead of synthetic insecticides in many pest control programs [13,14,15].

Plants and their derivatives have acquired great importance for humanity due to their different pharmacological properties and have been used in curing and treating disease; moreover, they have numerous other applications in different fields. Terpenes are volatile compounds that do not dissolve in water because they are hydrophobic [16]. The nano-encapsulation protocol for suitable biopolymers, lipid nanoparticles, and PLGA nanoparticles is one of the best ways to solve this problem, making it easy for natural oils to spread in water. In the presence of an emulsifying agent, oil and water are also mixed to create a nanoemulsion. Due to its small size (less than 100 nm) and good physicochemical properties, nanoemulsion is one of the most useful and widely used nano-systems [17,18,19]. Due to their numerous benefits and applications, essential oils and their nano-formulations have recently received substantial attention [16].

Among the important aromatic and therapeutic plants selected for our study are Mediterranean aloe (*Aloe vera* L.), bitter orange (*Citrus aurantium* L.), camel grass (*Cymbopogon schoenathus* L.), wild strawberry (*Fragaria virginiana* D.), lettuce (*Lactuca sativa* L.), English lavender (*Lavandula angustifolia* M.), honeysuckle (*Lonicera caprifolium* L.), common balm (*Melissa officinalis* L.), cream narcissus (*Narcissus tazetta* L.), sweet marjoram (*Origanum majorana* L.), scented geranium (*Pelargonium graveolens* L’Hér.), frangipani (*Plumeria rubra* L.), patchouli (*Pogostemon cablin* B.), pomegranate (*Punica granatum* L.), castor bean (*Ricinus communis* L.), and rosemary (*Rosmarinus officinalis* L.). In this context, we focus on honeysuckle and patchouli plants, as they are novel and promising plants, and compare the best results to other plants under study. Their effectiveness has not been previously studied on the West Nile vector *C. pipiens*.

One of the promising aromatic plants is honeysuckle (*Lonicera caprifolium*), which is found in North America and Eurasia, China, and southern Asia [20]. It is a perennial flowering plant used in traditional herbal medicine for its antiviral, antibacterial, and antioxidant activities and as a source of ecofriendly larvicides for mosquito control, and it is also used in pharmaceutical industries and medicinal and cosmetic preparations [21,22,23]. 

The other significant fragrant and therapeutic plant is patchouli (*Pogostemon cablin*). It is a perennial bushy herb or undershrub that originates in Southeast Asia, India, China, Indonesia, Malaysia, Thailand, and West Africa [24]. Patchouli oil has been used as a botanical insecticide on a limited scale against insect pests and possesses antibacterial, anti-inflammatory, and anticancer properties, and anti-insecticidal, insect repellant, antifungal, and antibacterial activities [25,26].

The focus is on *L. caprifolium* and *P. cablin* plants due to their high efficiency, and due to the lack of extensive information about their chemical compositions, little is known about their insecticidal capabilities. Therefore, our study supports the exploration of their chemical compositions and tests the larvicidal and adulticidal activity of two essential oils in the lab and field before and after their nano-formulation against the West Nile vector *C. pipiens*. The results of this study provide important information that could lead to the development of new mosquito-killing biopesticides that are safe for the environment.

## 2. Results

### 2.1. Screening of Larvicidal Activity for 16 Essential Oils

The larvicidal effects of 16 essential oils were screened against the late third-instar larvae of *C. pipiens* at 1500 ppm using the dipping technique. The results show that all plant oils had larvicidal activity (60–100% mortality, 24 h PT; 74.4–100% mortality at 48 h PT), and their lethal time (LT_50_) values ranged from 16.26 h (*L. sativa*) to 2.86 h and 3.01 h (*L. caprifolium* and *P. cablin*, respectively) (Table 1 and Figure 1). The efficacy of the oils could be classified based on the mortality rate (94–100%) at 24 h post-treatment (PT) as a highly effective group, which included four oils: *L. caprifolium*, *P. cablin*, and *R. officinalis* and *C. aurantium* at 100%, 100%, and 96%, respectively, 24 h PT (Table 1). The LT_50_ values of the highly effective oils were 2.86, 3.01, and 3.89 h for *L. caprifolium*, *P. cablin*, and *R. officinalis*, respectively, and the LT_95_ values were 13.56, 14.10, and 19.22 h, respectively. The relative effects (REs) of the highly effective group of oils according to LT_50_ values were 5.7, 5.4, and 4.2 times greater, respectively, than those of *L. sativa* (Figure 1).

The moderately effective oil group resulted in 80–94% mortalities at 24 h PT, including eight oils: *C. aurantium*, *C. schoenanthus*, *O. majorana*, *R. communis*, *L. angustifolia*, *P. graveolens*, *N. tazetta*, and *P. granatum*. At 24 h PT, these oils provided 84–94% mortalities (Table 1). The LT_50_ values of this moderate group ranged from 6.75 (*P. granatum*) to 4.12 h (*C. aurantium*), and their LT_95_ values ranged from 25.13 to 68.42 h, respectively (Figure 1). The REs for the LT_50_ values were 3.9 for *C. aurantium*, 3.6 for *Ricinus communis*, 3.5 for *C. schoenanthus*, 3.2 for *O. majorana*, 3.0 for *P. graveolens*, 2.8 for *L. angustifolia*, 2.7 for *N. tazetta*, and 2.4 for *P. granatum* (Figure 1).

The other five oils (*P. rubra*, *M. officinalis*, *A. vera*, *F. virginiana*, and *L. sativa*) were the least effective. The least effective ones were *Fragaria virginiana* and *L. sativa*, which killed 64 and 60% of the flies at 24 h PT, and their LT_50_ values were 11.15 and 16.26 h, respectively (Table 1 and Figure 1).

The results show that most of the oils used in this study exhibited a highly toxic effect, especially after 48 h of treatment, with mortality reaching 100% for *L. caprifolium*, *P. cablin*, *R. officinalis*, *C. aurantium*, *C. schoenanthus*, *O. majorana*, and *R. communis* and high levels for *N. tazetta* (97.6%), *P. graveolens* (96.8%), *L. angustifolia* (95.2%), and *P. rubra* (94.4%) oils (Table 1).

In addition to one-way analysis of variance (ANOVA) and Duncan’s multiple range tests, the Mann–Whitney U test was used to compare the mean differences across groups after the Kruskal–Wallis test to compare the mean differences of more than two groups. However, the Kruskal–Wallis and Friedman’s tests revealed that there were meaningful differences between the three groups at various points in time (Appendix A).

### 2.2. Larvicidal Activity of Two Effective Oils 

The larvicidal effects of oils and their nano-formulations were evaluated against the late third-instar larvae of *C. pipiens*. The mortality percentage post-treatment at 1500 ppm for 24 h with respect to honeysuckle and patchouli oils and nanoemulsions reached 100%. The LC_50_ and LC_95_ values were calculated for honeysuckle oil (247.72 and 1254.77 ppm, respectively), nano-honeysuckle (88.30 and 282.71 ppm, respectively), patchouli oil (276.29 and 1355.99 ppm, respectively), and nano-patchouli (93.05 and 321.52 ppm, respectively) (Table 2 and Table 3). 

Furthermore, the data show that nano-honeysuckle outperformed honeysuckle oil in terms of mortality, with 95.2% mortality at 250 ppm versus 36.8% mortality relative to the oil. Similarly, the data show that nano-patchouli was more effective than patchouli oil, with mortality rates of 91.2% and 34.4% at 250 ppm, respectively, after 24 h of treatment (Table 2 and Table 3). 

The results show that nano-honeysuckle (LC_50_ = 56.22 and LC_95_ = 134.4 ppm) and nano-patchouli (LC_50_ = 61.6 and LC_95_ = 158.42 ppm) were the most effective with respect to the lethal concentrations and had lower values compared to the honeysuckle and patchouli oils (130.63 and 605.47 ppm and 149.0 and 630.92 ppm, respectively) after 48 h of treatment (Table 4 and Table 5).

### 2.3. Efficiency of Essential Oils on Adult Mosquitoes

At 60 min of exposure, the effectiveness of the 16 essential oils against the adult mosquito *C. pipiens* was tested. The knockdown rate (K) and 50% knockdown time (KT_50_) values were determined. *C. pipiens* females were more susceptible to *C. schoenanthus* (KT_50_ = 58.59), *N. tazetta* (KT_50_ = 62.02), *P. graveolens* (KT_50_ = 64.34), *C. aurantium* (KT_50_ = 66.37), *L. caprifolium* (KT_50_ = 69.32), and *P. cablin* (KT_50_ = 72.77), with the knockdown percentage ranging from 17.78% to 55.55% at 1% concentration. The relative effects (REs) of the highly effective group of oils according to the LT_50_ values were 3.8, 3.6, 3.5, 3.4, 3.3, and 3.1 times greater, respectively, than *L. sativa* (Table 6).

At 5% concentration, *N. tazetta* (KT_50_ = 21.14), *L. caprifolium* (KT_50_ = 22.19), *P. cablin* (KT_50_ = 23.15), *C. schoenathus* (KT_50_ = 24.36), *C. aurantium* (KT_50_ = 26.56), *P. graveolens* (KT_50_ = 28.93), *L. angustifolia* (KT_50_ = 30.07), and *O. majorana* (KT_50_ = 36.36) had knockdown percentage values ranging from 37.78% to 93.33%. The relative effects (REs) of the highly effective group’s oils according to the LT_50_ values were 5.8, 5.5, 5.3, 5.0, 4.6, 4.2, 4.1, and 3.4 times greater, respectively, than that of *Aloe vera* (Table 7). The mortality percentages of the adults subjected to 5% concentrations of the 16 oils showed that *L. caprifolium* and *P. cablin* (100% mortality) were highly effective oils, followed by *N. tazetta* (97.78%), *L. angustifolia* (95.55%), and *R. officinalis* (95.56%), and then by *C. aurantium* and *P. graveolens* (93.33%) (Table 7). In contrast, *P. rubra* (80%), *F. virginiana* (75.56%), *L. sativa* (73.33%), and *A. vera* (71.11%) oils had lesser efficacy with respect to killing adult mosquitoes.

### 2.4. Adulticidal Activity for Two Effective Oils

The adulticidal effects of oils and their nano-formulations were evaluated against female mosquitoes. The data show that nano-honeysuckle and nano-patchouli were more effective than oils, with 100% mortality at 5% for 24 PT. The LT_50_ (50%, median time concentration) values were calculated for nano-honeysuckle and nano-patchouli (13.04 and 14.34 min, respectively) and the corresponding oils (22.93 and 23.93 min, respectively) (Table 8).

### 2.5. Characterization of Essential Oil Nanoemulsions 

Particle size distribution and polydispersity measurements

The particle size measurement in a nanoemulsion is a crucial parameter that confirms the particle distribution and motility, which reflect the nanoparticle’s stability and half-life [27]. The results of dynamic light scattering confirmed that the prepared nanoemulsions were within 100–300 nm, which is very good for drug delivery applications [28]. The DLS values of both honeysuckle and patchouli essential oils were 183 and 250 nm, respectively. Such comparable results are consistent with similar results obtained by similar studies [29,30]. For a more homogenous solution, the polydispersity index should not exceed 0.5 [31]. The results of the polydispersity index of the prepared nanoemulsions were 0.285 and 0.420 for honeysuckle and patchouli essential oils, respectively. For honeysuckle, the prepared nanoparticles presented some type of homogeneity, whereas the borderline polydispersity value of the patchouli nanoemulsion seemed to be completely broad (Figure 2 and Figure 3).

B.Charge distribution and stability (zeta potential)

The preparation of nanoemulsion is not truly the big dilemma, but keeping oil droplets far away from each other in the aqueous medium is; similarly, keeping the nanoemulsion system stable without aggregation or flocculation is challenging, which is the main monitor of the stability of the nanoemulsion system. The zeta potential is a measure of the charges of the particles and their stability in colloidal systems. The results obtained by the two nanoemulsion systems were 49.8 mV and 23.5 mV, respectively. The results presented by honeysuckle nanoemulsion were very interesting and expressed excellent stability. The zeta potential presented by the patchouli nanoemulsion did not differ from that of honeysuckle even though it had a lower negative charge compared to the honeysuckle nanoemulsion, and it still exhibited a very good stability profile (Figure 2 and Figure 3). 

C.Internal morphology via transmission electron microscopy (TEM)

Along with the particle size and charge mobility, the TEM analysis of the internal structure of the prepared nanoparticles is one of the most important factors. This is because it confirms the stability and checks for clusters. The prepared nanoemulsion of patchouli and honeysuckle furnished regular spherical and semispherical particles with widely variant particle sizes, as was previously proven by the higher polydispersity values. Figure 4 depicts a wide range of particle sizes ranging from less than 100 nm to 350 nm, as well as some agglomeration, particularly in the patchouli nanoemulsion.

### 2.6. Phytochemical Analysis of Honeysuckle and Patchouli Essential Oils 

Polyphenol content concentration determination using HPLC

Polyphenol identification of honeysuckle and patchouli was carried out via HPLC using 18 standards by comparing the standards to the essential oils; the identified components are presented in Table 9, and the structures of the tested polyphenol active components identified via HPLC are shown in Appendix A. The highly abundant polyphenols presented in the honeysuckle were vanillin, daidzein, and cinnamic acid, with concentrations of 8152.26, 3116.16, and 2447.91 µg/g, respectively, in addition to other good concentrations of syringic acid, gallic acid, quercetin, and catechin of 141.62, 70.95, 67.97, and 60.96 µg/g, respectively. Similar to honeysuckle, patchouli contained some important polyphenols, like apigenin, quercetin, gallic acid, daidzein, coumaric acid, and vanillin, with concentrations of 174.82, 170.25, 47.45, 18.66, 10.77, and 6.32 µg/g, respectively. At first glance, we can conclude that the honeysuckle essential oil was enriched with highly concentrated polyphenols compared to the patchouli essential oil (Figure 5).

b.GC–MS identification of volatile content

Phytochemical analysis was carried out using a GC–MS chromatogram for *L. caprifolium* and *P. cablin* (Table 10 and Table 11 and Appendix A) and some of the most effective oils as a second group, e.g., *C. aurantium*, *C. schoenanthus*, *L. angustifolia*, *N. tazetta*, *P. graveolens*, *O. majorana*, *P. granatum*, *R. communis*, and *R. officinalis* (Appendix A). These phytochemical compounds were identified by comparing their mass spectral fragmentation patterns, peak retention times, and peak areas (%) to those of known compounds listed in the National Institute of Standards and Technology (NIST) collection.

The phytochemical analysis of some of the most effective oils as a second group via GC–MS analysis revealed their major compounds. Greater abundance was found for limonene (91.35%) in *C. aurantium* oil; geranial (39.89%) and neral (38.04%) in *C. schoenanthus* oil; linalyl acetate (42.28%) and linalool (30.42%) in *L. angustifolia* oil; 7-acetyl-6-ethyl-1,1,4,4-tetramethyltetralin (17.98%), diethyl phthalate (13.35%), and methyl dihydrojasmonate (11.36%) in *N. tazetta* oil; terpinen-4-ol (23.50%), cis-4-thujanol (19.70%), and ℽ-terpinene (11.69%) in *O. majorana* oil; phenylethyl alcohol (35.51%) and 1-propanol, 2-(2-hydroxypropoxy) (15.77%) in *P. graveolens* oil; 14-β-H-pregna (42.39%) and diethyl phthalate (27.97%) in *P. granatum* oil; cyclobutane, 1,1-dimethyl-2-octyl (31.33%), and estragole (24.95%) in *R. communis* oil; and α-pinene (20.85%), eucalyptol (20.45%), camphor (14.65%), and borneol (11.32%) in *R. officinalis* oil (Appendix A, respectively). 

The identification of the chemical composition of *L. caprifolium* and *P. cablin* oils via GC–MS revealed 21 and 24 compounds representing 98.53% and 99.34% of the total oil, respectively. The chemical profiles from the analysis of the chromatograms of *L. caprifolium* and *P. cablin* varied in terms of qualitative and quantitative components. 

The phytochemical analysis of honeysuckle oil revealed 21 identified phytochemical compounds (Table 10), the most important of which were diethyl phthalate (24.85%), 7-acetyl-6-ethyl-1,1,4,4-tetramethyltetralin (21.69%), and β-methyl ionone (15.76%) as the major compounds (≥10%). In addition, 9,12-octadecadienoic acid (z, z) (5.49%), oxacycloheptadec-8-en-2-one, (8Z) (5.44%), ethylene brassylate (4.67%), α-methylionone (3.96%), β-ionone (3.06%), 5,5-dimethyl-2-(7-hydroxy-n-heptyl)-2-n-hexyl-1,3-dioxane (2.26%), 1-(4-isopropylphenyl)-2-methylpropyl acetate (1.54%), acetic acid, phenylmethyl ester (1.53%), citronellol (1.36%), linalyl acetate (1.25%), linalool (1.19%), and geraniol (1.17%) existed as minor compounds (≥1% and <10%), and the other compounds were traces (less than 1%).

The chemical constituents of patchouli oil were identified via GC–MS analysis (Table 11), indicating that the patchouli oil contained 24 identified chemical compounds. Patchouli alcohol (26.62%), α-bulnesene (16.88%), and α-guaiene (15.80%) were the major compounds. Seychellene (9.34%), 1H-3a,7-methanoazulene,2,3,6,7,8,8a-hexahydro-1,4,9,9-tetramethyl (1à,3aà,7à,8aá) (6.73%), α-patchoulene (3.80%), β-caryophyllene (3.44%), aciphyllene (3.31%), pogostole (2.58%), valencene (1.70%), and β-elemene (1.45%) were the minor compounds. The remaining compounds were present in trace amounts. 

### 2.7. Larvicidal Field Evaluation

Honeysuckle and patchouli oils and their nanoemulsions were utilized in LC_95_ X2 doses for the field evaluation of larvicides (2509.6 and 2711.9 ppm and 565.4 and 643.0 ppm, respectively), whereas only dechlorinated water was used at the control sites. The results show that there were not many larvae in the treated pools in the village of Kafr Saad PT. Treatments with honeysuckle and patchouli oils and their nanoemulsions reduced the larval density to 89.4, 86.5, 98.6, and 97.0%, respectively, at 24 h PT, with a persistence of five days for oils and eight days for nanoemulsions PT in pools (Figure 6). The Student’s T-test showed a significant difference between honeysuckle oil and nano-honeysuckle (F = 25.52, df = 3.84, *p* < 0.01); moreover, the T-test revealed a significant difference between nano-patchouli and patchouli oil (F = 27.23, df = 3.64, *p* < 0.01).

### 2.8. Adulticidal Field Evaluation

The adulticidal effects of the applied materials were evaluated against *C. pipiens* adults in some houses in Kafr Saad village PT, and the results show that nano-honeysuckle (100% mortality) was more effective than honeysuckle oil (98.0%); their LT_95_ values were 183.54 and 201.63 min, respectively (Figure 7). Furthermore, the data show that nano-patchouli was more persistent (7 days) than nano-honeysuckle (6 days). Similarly, the results show that patchouli oil (5 days) was more stable than honeysuckle oil (4 days) in the selected homes (Figure 7).

## 3. Discussion

*Culex* mosquito-borne diseases are a global concern. *C. pipiens* is a species that is known as a pest in urban environments. Female *C. pipiens* mosquitoes are the main vectors for the West Nile virus (WNV). They also transmit other arboviruses and act as a vector of filarial parasites, human lymphatic filariasis [32]. Due to the scarcity of treatment options, disease control relies on targeting the mosquitoes that transmit the disease in order to eliminate them. Therefore, it is important to use green pesticides, including essential oils, as environmentally friendly natural products for the manufacture of sustainable commercial pesticides. In order to achieve this, our study focused on studying 16 essential oils extracted from plants; then, we highlighted 2 essential oils, *L. caprifolium* and *P. cablin*, and their nanoemulsions, which produced the highest efficiency against *C. pipiens*.

Essential oils are natural chemical substances extracted from plants with different insecticidal chemical properties and different effective modes of action. They are used instead of synthetic or chemical pesticides to protect the environment and avoid the negative effects of synthetic pesticides [13]. Essential oils contain effective compounds that have the ability to combat mosquitoes by acting as insecticides, repellents, antifeedants, growth inhibitors, oviposition inhibitors, and ovicides, and they have growth-reducing effects that are larvicidal and toxic, thus reducing the abundance of disease vectors and the risk of disease [13,33]. These plant compounds are effective, safe, and biodegradable [30].

Preliminary screening demonstrated a wide range of toxicities among the 16 essential oils tested, and most oils demonstrated high toxicity at the screening concentration of 1500 ppm. *R. officinalis*, *C. aurantium*, *C. schoenanthus*, *O. majorana*, and *R. communis* were among the oils with a clear effect on *C. pipiens* larvae in the screening tests, though their effect was less than that of the honeysuckle and patchouli oils. In contrast, the data show that the *A. vera*, *F. virginiana*, and *L. sativa* oils were among the least toxic oils for *C. pipiens*.

Our data show that *N. tazetta* (KT_50_ = 20.94), *L. caprifolium* (KT_50_ = 22.89), *P. cablin* (KT_50_ = 23.45), *C. schoenathus* (KT_50_ = 24.36), *C. aurantium* (KT_50_ = 26.56), *P. graveolens* (KT_50_ = 28.93), *L. angustifolia* (KT_50_ = 30.07), and *O. majorana* (KT_50_ = 36.36) oils were highly effective when tested on female *C. pipiens* mosquitoes at 5% concentrations, and their relative effects according to their LT_50_ values were 5.8, 5.5, 5.3, 5.0, 4.6, 4.2, 4.1, and 3.5 times greater, respectively, than that of *Aloe vera*. Also, our data show that nano-honeysuckle and patchouli were more effective than their respective oils when tested individually, with 100% mortality at 5% at 24 h PT.

In general, there is a scarcity and significant lack of studies on the plants examined herein and their effect on *C. pipiens*. The following are the results from previous studies of essential oils extracted from the plants under study against *C. pipiens*. *C. aurantium* essential oil has a major limonene content (90%) and insecticidal activity on *C. pipiens* larvae. In addition, the essential oil showed 100% mortality with respect to larval stages 3 and 4 of *C. pipiens* at 300 ppm. The LC_50_ and LC_90_ values were 139.48 and 212.04, respectively [34,35]. Theochari et al. [36] found that limonene had larvicidal properties against two mosquito species, *Aedes albopictus* and *C. pipiens*. 

Likewise, *L. angustifolia* essential oil showed a toxic effect against *C. pipiens* larvae, with LC_50_ and LC_90_ values of 140 μg/mL and 450 μg/mL, respectively. Linalool, linalyl acetate, geraniol, lavandulyl acetate, camphor, β-caryophyllene, terpinen-4-ol, β-myrcene, and 1,8-cineole were the components of the essential oil [37,38]. In the same approach, *O. majorana* essential oil contained 4-terpinene (28.96%), γ-terpinene (18.57%), α-terpinene (12.72%), and sabinene (8.02%). Its LC_50_ and LC_90_ values against *C. pipiens* were 258.71 mg/L and 580.49 mg/L, respectively [39]. Regarding *P. graveolens*, Aboelhadid et al. [40] observed its insecticidal potency against the larvae of *C. pipiens*, and its LC_50_ value was 0.22%. 

On the same topic, *R. officinalis* essential oil exhibited larvicidal activity against the fourth-instar larvae of *C. pipiens* at 24 h (its LC_25_, LC_50_, and LC_90_ values were 39.47, 51.33, and 86.77 ppm, respectively). In addition, the oil contained 1,8-cineole (44.34%), camphor (16.74%), α-pinene (10.07%), and borneol (5.02%) as components [41]. Aouinty et al. [42] reported that the aqueous extract of *R. communis* leaves exhibited toxic action against the mosquito larvae of *C. pipens*, where significant histological changes, toxic action, hypertrophy, and the lysis of epithelium intestinal cells were observed, as well as musculature and external teguments of larvae and tissue lysis. With respect to *C. schoenanthus*, Wangrawa et al. [43] concluded that its essential oil exhibited larvicidal activity against *Anopheles funestus* and *C. quinquefasciatus*. Likewise, the essential oils of *P. granatum* and *R. communis* were shown to have larvicidal effects against mosquitoes [44,45].

Concerning *N. tazita*, methyl dihydrojasmolate is a major component and exhibited mosquito-repelling activity [46,47]. Also, methyl anthranilate, as one of the components of the essential oil, showed repellent properties against *Aedes aegypti* [48]. Limonene and α-pinene are effective inhibitors of acetylcholinesterase in various insects and larvae due to their ability to alter the activity of the insect’s acetylcholinesterase, which is required for neuro-neuronal and neuromuscular junctions in insects [49,50]. Terpineol is very effective in inhibiting insect reproduction, and the occurrence of monoterpenes and sesquiterpenes leads to fumigant toxicity [51]. 

Despite the superior properties of honeysuckle oil from our findings, honeysuckle essential oil has not been studied before or researched in terms of *C. pipiens*. There is a scarcity of research on honeysuckle and patchouli oils against mosquitoes. The only study was conducted on *L. caprifolium* essential oils by Muturi et al. [23], which proved its insecticidal property against *Aedes aegypti* and verified it as a source of ecofriendly larvicides for mosquito control. We also found one study conducted on patchouli essential oil and its larvicidal activity against *C. pipiens* [52]. 

Our work is somewhat consistent with or similar to the study of Hazarika [53] on *P. cablin* oil and its toxicity on *Aedes aegypti*. The essential oil was most effective in killing *A. aegypti* larvae, with an LC_50_ value of 25.14 mg/L. Similarly, the significant larvicidal potential of *P. cablin* essential oil on *A. aegypti* was reported, with high toxicity and an LC_50_ value of 24.25 μg·mL^−1^ [54]. Also, *P. cablin* oil showed repellent and insecticidal properties against *Aedes aegypti* [55]. The 16 μL/mL concentration of *P. cablin* essential oil displayed 100% repellency for *A. aegypti* [56].

The phytochemical analysis of *L. caprifolium* revealed diethyl phthalate (24.85%), 7-acetyl-6-ethyl-1,1,4,4-tetramethyltetralin (21.69%), and β-methylionone (15.76%) as the major compounds. In contrast, the main chemical compounds in *P. cablin* essential oil were patchouli alcohol (26.62%), α-bulnesene (16.88%), and α-guaiene (15.80%). Even though monoterpenes make up most of these essential oils’ compositions, we can observe from an analysis of their chemical makeup that these essential oils are substantially different. 

Few reports on the essential oil of *L. caprifolium* and its components have been published. Our results are consistent with the findings of Muturi et al. [23], who found that the major compounds that make up the essential oil are diethyl phthalate (0–23.2%), β-methylionone (0–1.1%), and 6-acetyl-1,1,2,4,4,7-hexamethyltetralin (0–23.19%). On the other hand, Ilies et al. [21] showed germacrene D (33.09%), farnesol (50.98%), nerolidol (2.63%), α-cadinol (2.86%), and linalool (1.93%) as the major compounds in *L. caprifolium*. In another study, linalool (16.42%), limonene (9.99%), and α-cadinol (10.65%) compounds were dominant in *L. caprifolium* essential oil [57]. 

For *P. cablin*, there are some published results on the essential oil and its composition that are similar to the results of our study. Lima Santos et al. [54] found that α-guaiene (13.3%), α-bulnesene (15.7%), and patchoulol (35.3%) are the major components. Moreover, patchouli alcohol may be related to biological activities. Likewise, patchouli alcohol (31.0%), α-bulnesene (21.3%), and α-guaiene 14.3%, are the same three major compounds [58]. Also, patchoulol (34.93%), α-bulnesene (17.76%), and α-guaiene (15.44%) were recorded as major compounds [59]. 

Variations in the composition of essential oils are influenced by different plant parts and their different stages of development and modifications due to the environment [60]. These factors influence the plant’s biosynthetic pathways and, consequently, the relative proportion of the main constituents. The insecticidal activity is probably due to the presence of monoterpenes, sesquiterpenes, and aromatic compounds with known biological properties [61]; moreover, various components of the essential oil and their roles in preventing or decreasing the activity of the enzyme acetylcholinesterase (AChE) and the effects on neurotransmitter receptors have been reported [62,63]. Insecticidal activity could be attributed to the action of specific compounds within the essential oil or the synergistic action of several molecules [64]. Furthermore, the diversity of oil components reduces insect resistance and behavioral habituation to deterrents. The toxic effects of essential oils can upregulate physiologically important proteins and enzymes in insects and synergize insecticide toxicity by inhibiting detoxification enzymes [62,63]. 

Our results indicate that *L. caprifolium* and *P. cablin* oils exhibited larvicidal and adulticidal activity, and their nanoemulsions showed increased efficacy and persistence. Honeysuckle and patchouli oils were found to be two of the most toxic essential oils to *C. pipiens* larvae and adults (100% mortality). Our data reveal that novel *L. caprifolium* oil effectively controlled the larvae and adults of *C. pipiens* (LC_50_ = 247.72 ppm and 22.13 min, respectively). The honeysuckle nanoemulsion was more effective than the crude oil against larvae and adults (LC_50_ = 88.30 ppm and LT_50_ = 13.04 min). Also, patchouli oil effectively controlled the larvae and adults of *C. pipiens* (LC_50_ = 276.29 ppm and LT_50_ = 22.93 min). The patchouli nanoemulsion was more effective than the crude oil against larvae and adults (LC_50_ = 93.05 ppm and LT_50_ = 23.93 min). 

There is a paucity of research published on *L. caprifolium* and *P. cablin* regarding the effects of their essential oil nanoemulsions. Furthermore, there are no previous studies on *L. caprifolium* and *P. cablin* essential oil nanoemulsions and their effects on *C. pipiens*. This study is, to our knowledge, the first research study on the effects of essential oil nanoemulsions on *C. pipiens*.

There are no reports on the effect of *L. caprifolium* and *P. cablin* essential oils on mosquitoes, especially *C. pipiens*, and there are no published studies on the effect of a nanoemulsion of *L. caprifolium* essential oil on any living organism.

Regarding *P. cablin*, Roshan et al. [59] demonstrated that a *P. cablin* essential oil-encapsulated chitosan nanoemulsion showed broad-spectrum antifungal and antimycotoxin activities and protected stored maize seeds from mold-induced biodeterioration and aflatoxin. Adhavan et al. [65] concluded that essential oil nanoemulsions from two wild patchouli species, *P. heyneanus* and *P. plectranthoides*, showed better antibacterial and anticandida activities in comparison with commercial patchouli essential oils. Another study evaluated the biological activity of pure patchouli essential oil compared with an emulsion containing 18% of the oil against tomato leafminer (*Tuta absoluta*) [66]. They added that the LD_50_ values were 10.06 and 2.57 µg of patchouli per mg of insect for the essential oil (EO) and emulsion, respectively. Oviposition was reduced in adults derived from the second instar treated with LD_10_ by 78.5% (EO) and 85.4% (emulsion) [66]. 

A nanoemulsion of *P. cablin* essential oil was more effective than the pure essential oil against adults of *Tetranychus urticae* and larvae of *Spodoptera litura*, and it displayed the highest efficacy in contact toxicity (LC_50_ 43.2 and 58.4 μg mL^−1^) after 48 h and fumigant toxicity (LC_50_ 9.3 and 13.6 μg mL^−1^) after 24 h against *Tetranychus urticae*. In addition, *P. cablin* nanoemulsion showed considerable antifeedant (AI) and feeding deterrent action (FI) (AI 99.21 ± 0.74 and FI 99.73 ± 1.24) against *Spodoptera litura* larvae [67]. 

There are some published reports on the effects of essential oils and their nanoemulsions on *C. pipens*. *Ocimum bascilicum* and *Cuminum cyminum* essential oils showed larvicidal effects on *C. pipiens*, with LC_50_ values of 81.07 ug/mL and 96.29 ug/mL, respectively. Also, nanoemulsion forms have higher efficiency, as evidenced by a reduction in the LC_50_ to 65.19 ug/mL for *O. bascilicum* and 64.50 ug/mL for *C. cyminum* [68]. Muturi et al. [69] demonstrated that *Commiphora erythraea* essential oil is a promising source of mosquito larvicide; LC_50_ values for the whole essential oil were 19.05 ppm for *C. restuans*, 22.61 ppm for *C. pipiens*, and 29.83 ppm for *Aedes aegypti*. Using *C. horaerythraea* essential oil emulsions enhanced the insecticidal properties of *C. erythraea* essential oil, and they were more toxic than the whole essential oil. In another study, *Cirtus sinensis* essential oil nanoemulsion was recommended for the control of vector-borne *C. pipiens* larvae disease; the LC_50_ values for the nanoemulsion and bulk emulsion were 27.4 and 86.3 ppm, respectively, and the larvicidal activity of the essential oil nanoemulsion was greater [70]. 

The goal has always been to search for environmentally friendly natural pesticides that are safer and more efficient than the synthetic pesticides that were once used to control mosquitoes, which are now seriously harming human health and, more significantly, breeding resistant mosquitoes. Phytochemicals have pesticidal effects and other harmful effects on mosquito physiology at different stages of development [71,72]. In addition, insects have a greater chance of developing resistance to a single chemical component compared to a range of chemicals. Therefore, the use of combined phytochemicals would prevent the emergence of mosquito resistance. Also, phytochemicals have a short residual half-life, and their use in combination with other biological control agents can be beneficial [73,74]. It is promising and encouraging that these phytochemical properties are present in essential oils. However, because they are volatile substances, their long-term applications in mosquito control have problems. The effectiveness and duration of essential oils have increased in recent years thanks to new technologies like microencapsulation and nanoemulsion [31,75].

The mixture of two phases of water and oil is called an emulsion; if the droplet size is at the nanoscale, it is called a nanoemulsion [76]. Emulsions are divided into two types: oil-in-water (O/W) and water-in-oil (W/O); in the former, oil droplets are dispersed in water, whereas in the latter, the opposite is true. In both cases, surfactants or surface tension-reducing agents are used to mix the two phases. In particular, O/W nanoemulsions are much more frequently used because most drugs and all essential oils are lipophilic and should be solubilized in blood flow or water [77]. 

In a similar study, Radwan et al. [78] discovered that nano-clays intercalated with green tea and fennel, Mg-LDH-GT, and Mg-LDH-F were the best-loaded systems, with relatively good desorption release to their active ingredients, and significantly affected *C. pipiens* larvae and adults in both laboratory and field conditions; moreover, the stability of the oil was improved.

Surfactants are crucial components of nanoemulsions. Surfactants come in four different varieties: cationic, anionic, amphoteric, and nonionic. Nonionic surfactants are typically encapsulated in the nanoemulsion when creating nanoemulsion-based pesticide applications because they are less sensitive to pH and ionic strength. Due to the cohesiveness between the anionic surfactant and the solution, this extra component has the potential to change the stability and size of the nanoemulsion [79]. The most used pesticide formulations consist of active compounds that can kill weeds, insects such as mosquitoes and ticks, and other organisms such as snails, slugs, and fungi (fungicides). The nanoemulsion acts as a “carrier” that carries and delivers bioactive ingredients to pests on plants. This is carried out to obtain the most effective pesticide delivery [80].

Our data obtained via dynamic light scattering confirmed that oil nanoemulsions were within 100–300 nm, whereas the DLS values of both honeysuckle and patchouli essential oils were 183 and 250 nm, respectively. Also, data show nanoparticle homogeneity for honeysuckle and patchouli via the polydispersity value, which indicates greater particle stability [81,82]

Our findings are consistent with those of Nuchuchua et al. [83], who studied citronella (*Cymbopogon nardus*), hairy basil (*Ocimum Americanum*), and vetiver (*Vetiveria zizanioides*) oil nanoemulsions against *A. aegypti*; nanoemulsions were prepared using a high-pressure homogenization technique and had mean droplet sizes ranging from 150 to 220 nm. This result could prolong mosquito protection times by up to 4.7 h due to the higher release rate and better physical stability of the nanoemulsion [83].

Copaiba (*Copaifera duckei*) oleoresin oil-in-water nanoemulsions had the lowest concentration at 200 ppm. The nanoemulsion oil killed 70% of *A. aegypti* larvae after 24 h and 90% after 48 h. The average size of the droplets was 145.2 nm [84]. The essential oil of sweet oranges (*Cirtus sinensis*) was used to create a larvicidal nanoemulsion with a mean droplet size of 78.8 nm and a polydispersity index (PDI) of 0.28. After 24 h against *C. pipiens*, the nanoemulsion’s LC_50_ value was 27.4 ppm and that of the bulk emulsion was 86.3 ppm [85].

Our data indicate, via the analysis of the studied oils using HPLC to determine the concentration of polyphenol contents, the presence of polyphenols in honeysuckle and patchouli oils. Polyphenols are abundant in honeysuckle oil through multiple organic compounds, such as vanillin, daidzein, and cinnamic acid, in addition to good concentrations of others, such as syringic acid, gallic acid, quercetin, catechin, methyl gallate, and naringenin. Patchouli oil also contains polyphenols such as gallic acid, coumaric acid, and vanillin.

Turgut et al. [86] reported that gallic, protocatechuic, p-hydroxybenzoic, vanillic, caffeic, chlorogenic, syringic, p-coumaric, ferulic, o-coumaric, rosmarinic, and trans-cinnamic acids are the phenolic acids found in *L. caprifolium*. Phenolic acids such as benzoic acid, cinnamic acid, vanillic acid, salicylic acid, p-hydroxybenzoic acid (p-HBA), and tetradecanoic acid were isolated and identified from *P. cablin* by Wu et al. [87]. Also, Wang et al. [88] identified apigenin and acacetin from *P. cablin*. Moreover, some reports showed the ovicidal, insecticidal, larvicidal, and acaricidal agents of polyphenols, e.g., p-coumaric, dihydrocoumaric, ferulic (E)-cinnamic, hydroxycinnamic, gallic, and caffeic acids [89,90].

Our data agree with recent findings with respect to Italian honeysuckle essential oil, which showed that the whole essential oil (LC_50_ of 34.4 mg/L) was two times less toxic to *A. aegypti* larvae than four of its five fractions (LC_50_ = 20.6, 19.7, 18.6, and 17.7 mg/L for fractions B, C, D, and E, respectively) and some *L. scoparium* essential oils. Similar results were also shown to be true for the essential oil of parsley (*Petroselinum crispum*), where the LC_50_ values for fractions 1, 3, and 4 against *A. aegypti* larvae were 0.49, 0.88, and 0.01 mg/L, respectively, as opposed to 4.19 mg/L for the entire essential oil [91].

Patchouli essential oil-loaded hydrogel activities may be due to the synergistic effects of the present associated compounds, such as patchoulene, patchoulol, pogostol, and patchouli alcohol, which has been reported in the literature to have potential anti-inflammatory and antimicrobial activity [80].

With almost half of the doses applied in the field, the larval reduction percentage of honeysuckle and nanoemulsion reduced larval densities (91% and 98.6%, respectively, 24 h PT), and their effect lasted (reduction % > 50%) for five and eight days PT, respectively, in pools. In ditches, reduced larval density reached 90% and 95.0% at 24 h PT, and the effect lasted (reduction % > 50%) for 4 and 6 days PT, respectively. Those of patchouli oil and nanoemulsion were 89.0 and 96.0% 24 h PT and persisted for six days 8 days PT, respectively. Similarly, the reduction was 91% and 98.6%, 24 h PT and their effect lasted (reduction % > 50%) for five and eight days PT, respectively, in ditches.

On the other hand, the reduction in adult density PT with respect to nano-honeysuckle and its oil and nano-patchouli and its oil reached 84.00, 100, 77.4, and 98.00%, respectively, and it was effective for five and three days relative to the nanoemulsions and oils. To the best of our knowledge, there was no previously filed application for the use of *L. caprifolium* and *P. cablin* oil nanoparticles against mosquitoes. Our search demonstrates that both *L. caprifolium* and *P. cablin* oils are highly effective essential oils that can be used against *C. pipiens*. 

Once more, some essential oils have the ability to repel a variety of insects, including vectors [56]. Some insects were subjected to the neurotoxic effects of essential oils, which led to hyperactivity, followed by hyperexcitation, and they were quickly knocked down and rendered immobile [58]; moreover, suffocation occurred due to the detrimental effects of bioactive plant products on neurotransmitter receptors [92]. Bioactive plant products, according to Baz et al. [70], have anti-insect effects that are used to eradicate insects, such as feeding deterrents/antifeedants, toxicants, growth retardants, repellents, chemosterilants, and attractants [93]. 

Many pesticides that are widely used in agricultural applications are highly hydrophobic particles with low water solubility; therefore, they must be loaded within appropriate delivery systems prior to use. The advantages of nano-formulations include the increased apparent solubility of poorly soluble active ingredients, slow/targeted release of the active ingredient, and protection against premature degradation. By inserting essential oils into the structure of nanoemulsions, the oil’s stability and ability to resist pests are high [94].

## 4. Materials and Methods

### 4.1. Chemicals and Plant Oils 

Sixteen essential oils were purchased from the Nefertari Company (Fayoum, Cairo, Egypt) for natural plant oils and cosmetics. The oils were 100% pure and natural and were extracted from natural plants via hydrodistillation (Table 12; Figure 8). De-ionized water, butanol, polysorbate 20, sodium glycocholate, and oleic acid were purchased from Alfa Esar (Thermo Fisher GmbH, Kandel, Germany) and used without further purification.

### 4.2. Mosquito Colony

*C. pipiens* mosquito larvae were used for all investigations and have been reared and maintained in an insectary at the Department of Entomology, Faculty of Science, Benha University, for many generations (F12). Mosquito larvae were reared in enamel plates (30 × 25 × 15 cm), filled with 2 L of stored dechlorinated tap water, and given powdered dog biscuits and Tetramin^®^ fish food (*w*/*w*) every 2 days. The colony was kept in good condition at 27 ± 2 °C, 75–80% RH, and a 12:12 h (L/D) photoperiod. Pupae were transferred to a mosquito cage (30 cm × 25 cm × 25 cm). Mosquito adults received an 8% sucrose solution as food. Both larvae and adults were kept in identical laboratory settings and were continually available for the tests [95].

### 4.3. Larvicidal Efficacy In Vitro

Sixteen oils were screened for their larvicidal efficacy [94] against the last third-instar larvae of *C. pipiens*. Oils were added to a solvent consisting of dechlorinated water plus 0.5 mL of 0.5% Tween 20 (an emulsifier) using a shaker plate to yield a homogenous solution [14]. Twenty larvae were placed in a 250 mL glass beaker containing 200 mL of 1000 ppm. Larval mortality for each oil concentration was recorded 24 h post-treatment (PT). After determining which oils were the most preferred for controlling mosquito larvae, two oils were chosen (mortality more than 95%), and different concentrations of two of the essential oils and their nanoemulsion formulations (50, 100, 250, 500, 1000, and 1500 ppm) were tested according to the WHO’s guidelines [96]. The control group was treated with the solvent only. The experiment was replicated five times. For two selected essential oils and their nanoemulsions, the larval mortality for each oil concentration was recorded at 24 h and 48 h post-treatment (PT).

### 4.4. Adulticidal Efficacy In Vitro

Using CDC bottle bioassays, adult mosquito susceptibility testing for the promising oils was carried out based on Vatandoost et al. [97]. Oils were placed in three bottles, with one for each concentration. Using pure ethanol as a solvent, different concentrations of each oil (0.5, 1, 2, 4, and 5%) were created. The requisite concentrations were applied to the bottles, and they were left to evaporate for 20 min at 28 ± 2 °C. To fill each bottle, a hand aspirator was used to remove 15 adult mosquitoes (aged 3–4 days) from the cage. The mosquitoes were taken out of the bottles after one hour of exposure and placed in separate paper cups with a 10% sucrose solution. Mortality was then calculated after 24 h using three replicates.

### 4.5. Preparation of the Essential Oil Nanoemulsion 

The nanoemulsion preparation protocol for essential oil was carried out using the homogenization method by Radwan et al. [33], with equivalent amounts of the essential oil and oleic acid, as follows: In a 50 mL beaker (B1), amounts of 2.5 mL of essential oil and 2.5 mL of oleic acid were mixed and warmed to 45 °C; in another 50 mL beaker (B2), 10 mL of distilled water, 0.2 g of sodium glycocholate, 0.25 mL of butanol, and 3 mL of polysorbate 20 were added and mixed very well using a hotplate stirrer until the final solution became homogenous. The temperature was monitored using an infrared thermometer; 45 °C was reached, and the same temperature was retained. To obtain the primary emulsion, the contents of the two beakers, B1 and B2, were mixed together at the same temperature under stirring, and the mixture was quenched rapidly with the addition of ice-cold water to obtain a final volume of 40 mL, which was reserved in a 50 mL Falcon tube (Figure 9).

### 4.6. Phytochemical Analysis 

Polyphenol content concentration determination via HPLC

High-performance liquid chromatography for polyphenol detection analyses was accomplished using the Agilent 1260 series on both honeysuckle and patchouli oils. The separation process was carried out using an Eclipse C18 column (4.6 mm × 250 mm i.d., 5 μm). The mobile phase consisted of two solutions, a mixture of water (A) and 0.05% trifluoroacetic acid in acetonitrile (B), injected at a flow rate of 0.9 mL/min. The mobile phase was programmed consecutively in a linear gradient as follows: 0 min (82% A), 0–5 min (80% A), 5–8 min (60% A), 8–12 min (60% A), 12–15 min (82% A), 15–16 min (82% A), and 16–20 min (82% A). The multi-wavelength detector was adjusted for a wavelength of 280 nm. For each sample solution, a volume of 5 μL was injected. The column temperature was adjusted to be maintained at 40 °C. 

b.Volatile content identification via GC–MS

Analyses of the selected essential oils were performed via GC–MS (gas chromatography–mass spectrometry). Thermo Scientific Trace GC Ultra/ISQ Single Quadrupole MS and TG-5MS fused silica capillary columns at 0.1 mm, 0.251 mm, and 30 m thick were utilized for GC–MS, which was employed for biochemical analyses. This was achieved using an electronic ionizer with 70 eV of ionization energy. As a carrier gas, helium was used (flow rate: 1 mL/min). The MS transmission line and injector were both set to 280 °C. The oven was preheated to 50 °C and then increased to 150 °C at a rate of 7 °C per minute, 270 °C at a rate of 5 °C per minute (pause for two minutes), and finally 310 °C at a rate of 3.5 °C per minute (continued for 10 min). A relative peak area was employed to explore the quantification of all components discovered. The chemicals were at least partially identified by comparing the retention times and mass spectra of the chemicals to those of NIST and Willy Library data from the GC–MS instrument. Identification was carried out using the aggregate spectrum of user-generated reference libraries. To evaluate peak homogeneity, single-ion chromatographic reconstructions were performed. To verify GC retention periods, co-chromatographic analyses of reference substances were used whenever practical [98].

### 4.7. Characterization of Essential Oil Nanoemulsion

Average droplet size and surface charge

The droplet size, radius, and polydispersity index (PDI) were measured using the dynamic light scattering (DLS) technique. The measuring conditions were set at room temperature with an angle of 173°. The net surface charge, or zeta potential (z.p.), was monitored by measuring the change in the frequency shift of the scattered light at a scattering angle of 12° due to laser beam irradiation. The average size, PDI, and zeta potential measurements were carried out using Zetasizer Nano ZS (Malvern Instruments Ltd., Malvern, UK) at the Egyptian Petroleum Research Institute (EPRI). The sample was prepared by dispersing about 5–10 mg of the solution under investigation in 10 mL of distilled water at a temperature of 25 °C, with homogenization for two minutes before measurements.

b.Nanoemulsion droplet morphology via Transmission Electron Microscopy (TEM)

Visualization of the shape of the nanoemulsion droplets and their internal structure was carried out via field transmission microscopy (HR-TEM and JSM-7100F) in the central labs of the Egyptian Petroleum Research Institute (EPRI), Cairo. Images were taken using a JEOL JEM-2100-115 high-resolution transmission electron microscope system with an accelerating voltage that varied from 100 to 200 kV. The sample was prepared by diluting 1 µL of N.E. with distilled water at a dilution factor of 1:200 and placing it on a 200-mesh carbon-coated grid for 2 min before removing the excess liquid via absorption through a cellulose filter. One to two drops of 2% (*w*/*w*) phosphotungstic acid (PTA) were dropped onto the grid for 10 s to achieve negative staining, and the excess PTA was disposed of via filter paper by absorption.

### 4.8. Larvicidal Field Evaluation of Patchouli and Honeysuckle Nanoemulsions

In November 2022, honeysuckle and patchouli oils and their nanoemulsions were tested on mosquito larvae in small, still-water ditches in Kafr Saad village, Egypt (283304200 N, 335605700 E, altitude 2624). The ditches averaged 75 to 120 m in length, with 1.30 m^2^ water surfaces and a depth of 0.55 m. The larval breeding ditches were selected to be adjacent to the population, contain steady water, and have high mosquito stage density. The honeysuckle and patchouli oils and their nanoemulsions (500 mL/m^2^) at a dose of LC_95_ X2 (8932.68, 1659.34, 11,729.2 ppm, and 2251.24 ppm, respectively) were applied to the breeding sites [31,68]. The essential oils were dissolved in Tween 20 prior to the application to ditches (about 400 mL of oil were mixed with 0.2 mL of a solution of 0.05% Tween 20 *v*/*v*). For each treatment, three replicates were performed. Before and after treatment, mosquito larvae were sampled from each site every day for a week. To examine the efficacy of the selected larvicides on the mosquito population, larval instars were collected from field water at each site using an enamel pad (450 mL), and each larvicide was treated and transported to the laboratory to count the mosquito larvae in order to determine mortality; live larvae were counted until adulthood to determine the persistence of the tested materials [99].

### 4.9. Adulticidal Field Evaluation of Patchouli and Honeysuckle Nanoemulsions

The efficacy and stability testing of honeysuckle and patchouli oils and their nanoemulsions on adult mosquitoes was carried out in some homes in Kafr Saad village, and these homes contained humans and sometimes their animals. According to the guidelines of the WHO [71], honeysuckle and patchouli oils and their nanoemulsions (LC_95_ X2) were sprayed into three rooms in each selected home for 5 min; then, the room’s door was closed. Three rooms were sprayed with dechlorinated water as a control group. The reduction in adult mosquitoes was calculated according to the WHO’s guidelines [99]. Before spraying the tested materials in the rooms, white cloth was spread on the surface of the room’s floor to collect dead mosquitoes after spraying in order to determine the density of adult mosquitoes; in untreated rooms, the adult mosquitoes were collected using CDC light traps, and the nets of the traps were placed in the freezer (20 °C) for 10 min to count the mosquitoes.

### 4.10. Data Analyses

The data were analyzed using the software SPSS V23 (IBM, Armonk, NY, USA), and probit analyses were carried out to calculate the lethal concentration (LC) values and a one-way analysis of variance (ANOVA) (Duncan’s MRT). The significance level was set at *p* < 0.05. The non-parametric Kruskal–Wallis test was performed to compare the mean differences of more than two groups, followed by the Mann–Whitney U test to compare the mean differences between the active oil groups. The relative efficacies (REs) were calculated according to the following formula [13]:RE for LC = LC_50_ (LC_90_ or LC_99_) for reference oil/LC_50_ (LC_90_ or LC_99_) for EO.

## 5. Conclusions

This study revealed for the first time the efficacy of honeysuckle and patchouli essential oils against the larvae and adults of *C. pipiens* mosquitoes due to the large diversity and high efficacy of the phytochemical compounds that are found in honeysuckle and patchouli essential oils. The use of these natural oils as environmentally benign pesticides is a direction that must be followed, and our research shows the potential of these oils against insects that pose a threat to human health. Therefore, we recommend the use of honeysuckle oil in mosquito vector management as a green insecticide because it is inexpensive, safe, environmentally friendly, and highly efficient. Finally, further studies can be directed toward the effect of these oils against non-target organisms.

## Figures and Tables

**Figure 1 plants-12-03682-f001:**
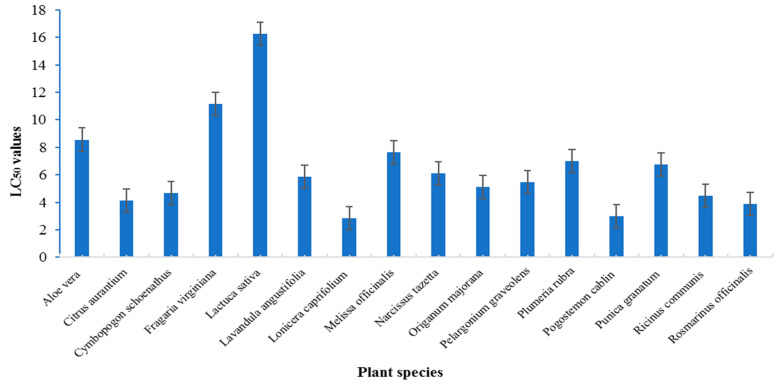
Lethal time values of oils applied at 1500 ppm against *Culex pipiens* larvae.

**Figure 2 plants-12-03682-f002:**
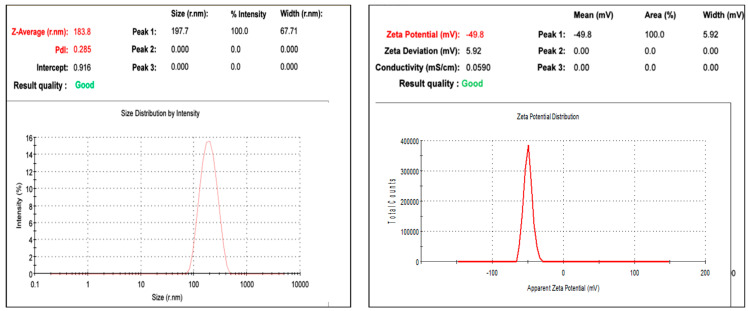
Mean particle size, polydispersity index (PDI), and zeta potential of the honeysuckle EO nanoemulsion.

**Figure 3 plants-12-03682-f003:**
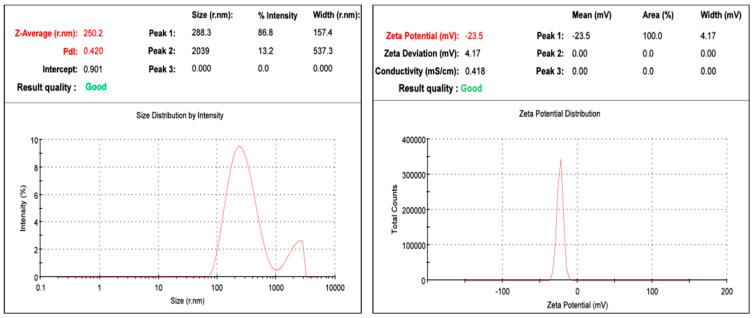
Mean particle size, polydispersity index (PDI), and zeta potential of the patchouli EO nanoemulsion.

**Figure 4 plants-12-03682-f004:**
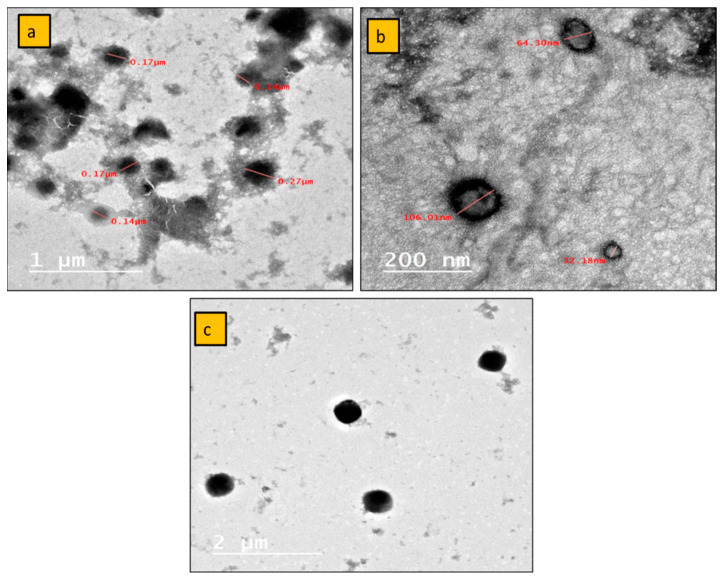
TEM morphological examination of patchouli (**a**,**b**) and honeysuckle (**c**) nanoemulsions.

**Figure 5 plants-12-03682-f005:**
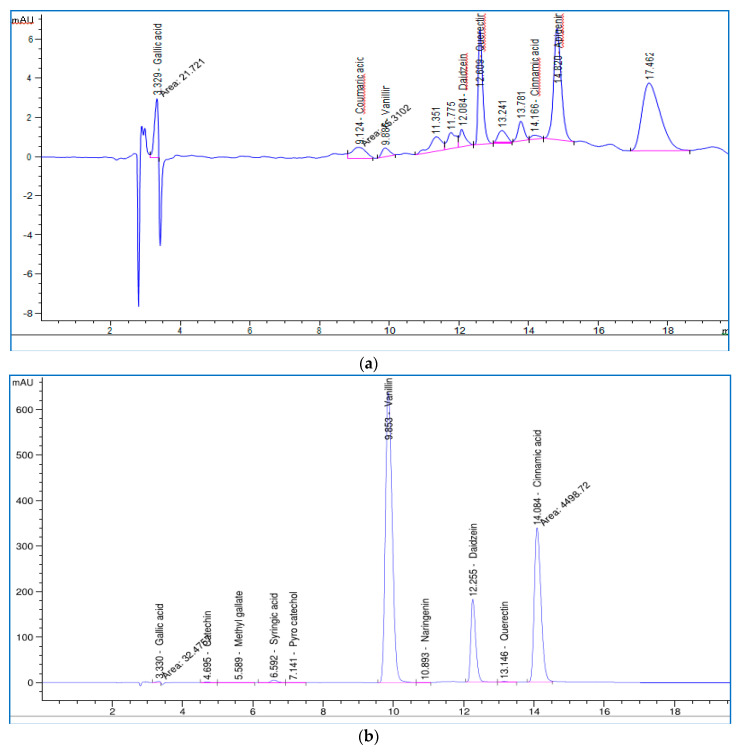
Chromatogram of the polyphenol concentration determination of patchouli (**a**) and honeysuckle (**b**) essential oils and standard polyphenols (**c**).

**Figure 6 plants-12-03682-f006:**
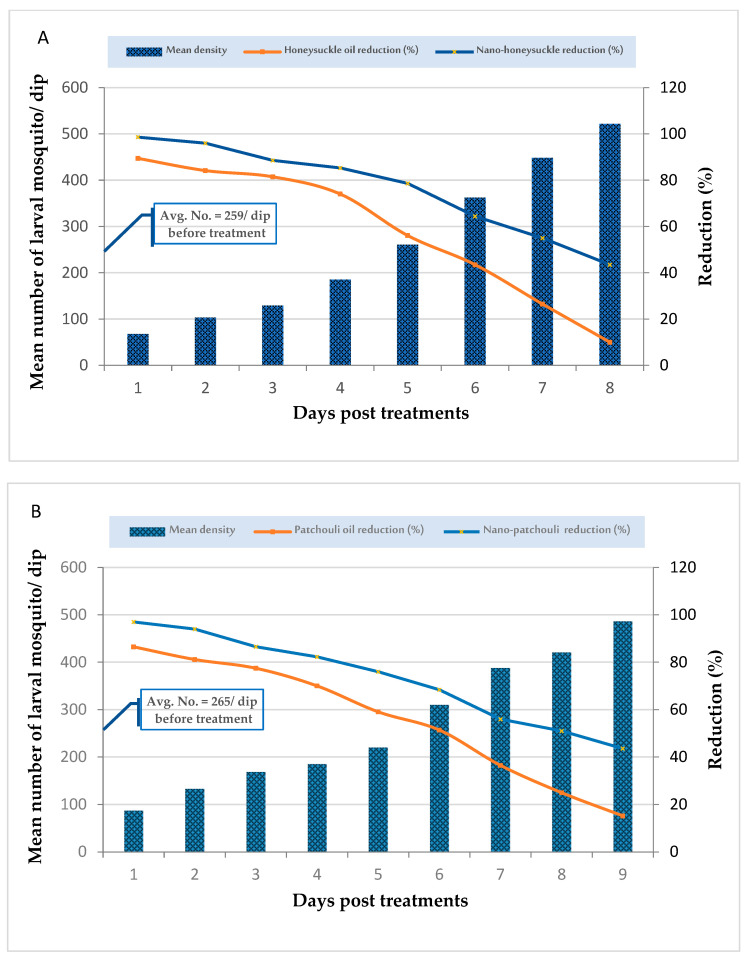
Field efficacy of nano-honeysuckle and its oil (**A**) and nano-patchouli and its oil (**B**) treated at a dose of LC_95_ X2 (2509.5, 565.4, 2711.9, and 643.0 ppm, respectively) in larval breeding sites.

**Figure 7 plants-12-03682-f007:**
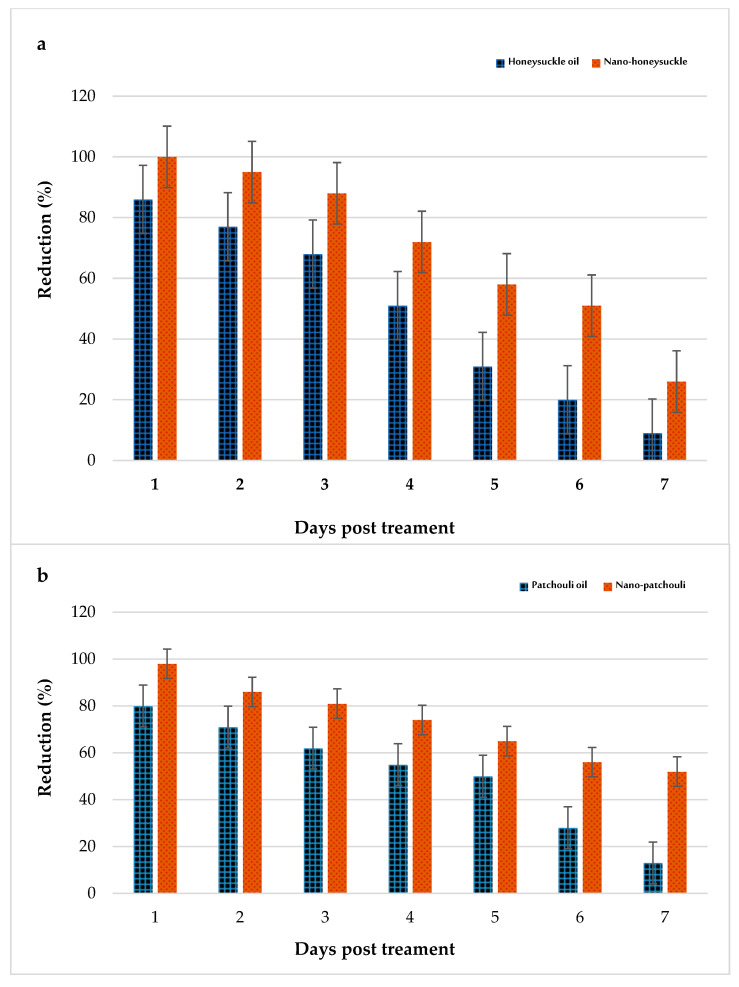
Persistence of honeysuckle oil and its nanoemulsion (**a**) and patchouli oil and its nanoemulsion (**b**) against adult mosquitoes in treated houses 30 min post-exposure for 7 days.

**Figure 8 plants-12-03682-f008:**
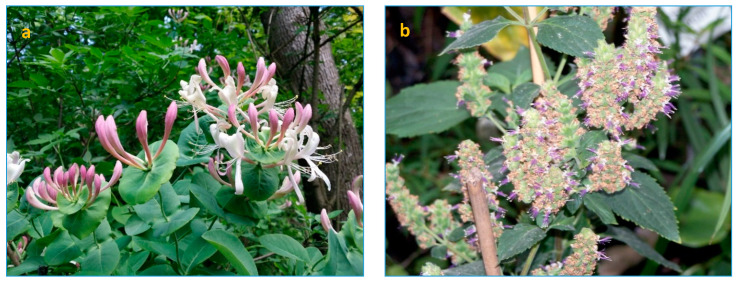
Honeysuckle, *Lonicera caprifolium* (**a**), and patchouli, *Pogostemon cablin* (**b**) plants. (https://upload.wikimedia.org).

**Figure 9 plants-12-03682-f009:**
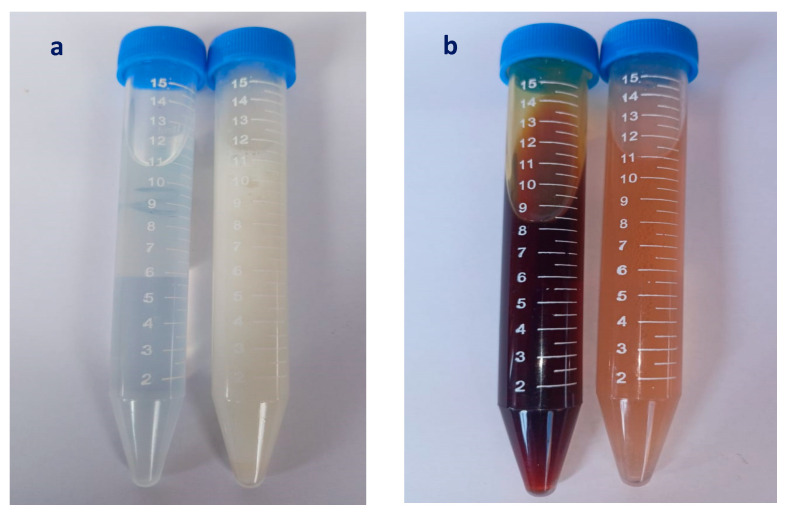
Honeysuckle (**a**) and patchouli (**b**) nanoemulsions.

**Table 1 plants-12-03682-t001:** Larval mortality (%) of plant oils used at 1500 ppm through different time periods against the 3rd larval mosquito *Culex pipiens*.

Oil Name	Mortality % (Mean ± SD)/h					
	0.0	3	6	12	24	48
*Aloe vera*	0 ± 0 ^aF^	28.8 ± 4.45 ^hE^	42.4 ± 5.88 ^gD^	60.0 ± 4.0 ^iC^	72.0 ± 2.19 ^hB^	86.4 ± 3.25 ^eA^
*Citrus aurantium*	0 ± 0 ^aF^	40.0 ± 2.83 ^bcE^	64.8 ± 3.44 ^cD^	84.0 ± 2.53 ^cC^	94.4 ± 3.49 ^bcB^	100 ± 0.00 ^aA^
*Cymbopogon schoenathus*	0 ± 0 ^aF^	38.4 ± 2.99 ^cdE^	62.4 ± 5.15 ^cdD^	82.4 ± 2.71 ^cdC^	92.0 ± 2.53 ^cB^	100 ± 0.00 ^aA^
*Fragaria virginiana*	0 ± 0 ^aF^	23.2 ± 2.33 ^iE^	36.8 ± 3.44 ^hD^	51.2 ± 3.44 ^jC^	64.0 ± 3.35 ^iB^	84.8 ± 1.50 ^eA^
*Lactuca sativa*	0 ± 0 ^aF^	15.2 ± 1.50 ^jE^	27.2 ± 3.44 ^iD^	42.4 ± 1.60 ^kC^	60.0 ± 2.19 ^jB^	74.4 ± 3.49 ^fA^
*Lavandula angustifolia*	0 ± 0 ^aF^	35.2 ± 4.08 ^defE^	56.8 ± 5.28 ^eD^	80.0 ± 3.79 ^deC^	88.0 ± 3.35 ^deB^	95.2 ± 2.94 ^bcA^
*Lonicera caprifolium*	0 ± 0 ^aF^	52.8 ± 3.88 ^agD^	79.2 ± 3.44 ^aC^	94.4 ± 3.92 ^aB^	100 ± 0.00 ^aA^	100 ± 0.00 ^aA^
*Melissa officinalis*	0 ± 0 ^aF^	29.6 ± 3.49 ^ghE^	47.2 ± 2.94 ^fD^	64.8 ± 1.50 ^hC^	76.8 ± 3.88 ^gB^	92.0 ± 2.83 ^cdA^
*Narcissus tazetta*	0 ± 0 ^aF^	32.8 ± 5.57 ^fE^	50.4 ± 7.44 ^fD^	75.2 ± 2.94 ^fC^	85.6 ± 3.49 ^efB^	97.6 ± 2.40 ^abA^
*Origanum majorana*	0 ± 0 ^aF^	33.6 ± 2.99 ^efE^	60.0 ± 2.53 ^deD^	80.8 ± 2.65 ^deC^	91.2 ± 2.94 ^cdB^	100 ± 0.00 ^aA^
*Pelargonium graveolens*	0 ± 0 ^aF^	35.2 ± 3.20 ^defE^	58.4 ± 2.71 ^eD^	78.4 ± 2.04 ^efC^	86.4 ± 3.92 ^efB^	96.8 ± 2.33 ^abA^
*Plumeria rubra*	0 ± 0 ^aF^	29.6 ± 5.15 ^ghE^	47.2 ± 3.88 ^fD^	64.8 ± 3.44 ^hC^	78.4 ± 4.12 ^fB^	94.4 ± 2.71 ^bcA^
*Pogostemon cablin*	0 ± 0 ^aF^	50.4 ± 3.49 ^aE^	76.8 ± 1.50 ^aD^	92.0 ± 3.79 ^aC^	100 ± 0.00 ^aA^	100 ± 0.00 ^aA^
*Punica granatum*	0 ± 0 ^aF^	29.6 ± 3.71 ^ghE^	48.8 ± 2.33 ^fD^	68.8 ± 2.94 ^gC^	84.0 ± 3.20 ^fB^	90.4 ± 2.71 ^dA^
*Ricinus communis*	0 ± 0 ^aF^	36.8 ± 4.96 ^cdeE^	62.4 ± 2.71 ^cdD^	82.4 ± 3.49 ^cdC^	90.4 ± 3.49 ^cB^	100 ± 0.00 ^aA^
*Rosmarinus officinalis*	0 ± 0 ^aF^	42.4 ± 2.40 ^bE^	68.8 ± 4.63 ^bD^	88.0 ± 4.20 ^bC^	96.0 ± 3.10 ^bB^	100 ± 0.00 ^aA^
Control	0 ± 03 _aA_	0 ± 0 ^kA^	0 ± 0 ^jA^	0 ± 0 ^lA^	0 ± 0 ^kA^	0 ± 0 ^gA^

Numbers in the same row followed by the capital letters (A–F) and numbers in the same column followed by the same small letters (a–l) are not significantly different (one-way ANOVA, Duncan’s MRT, *p* > 0.05); H: the highly effective group (95–100% mortalities), 3 oils; M: the moderately effective group (80–94% mortalities), 8 oils; L.: the least effective group, including the rest of the oils, 5 oils, 24 h post-treatment.

**Table 2 plants-12-03682-t002:** Efficacy of larvicidal activity of honeysuckle oil and its nanoemulsion against *Culex pipiens* 24 h post-treatment.

Oil Name	Tested Materials	Conc. (ppm)	Mortality (%)	LC_50_ (Low–Up.)	LC_90_ (Low–Up.)	LC_95_ (Low–Up.)	Slope	Chi (Sig.)
Honeysuckle (*Lonicera caprifolium*)	Oil	Control	00.0 ± 00 ^f^	247.72 (132.64–422.95)	876.88 (675.72–425.32)	1254.77 (1046.65–4075.57)	2.334 ± 0.150	31.225 (0.000)
		50	10.4 ± 2.71 ^e^					
		100	20.8 ± 1.50 ^d^					
		250	36.8 ± 2.33 ^c^					
		500	65.6 ± 2.71 ^b^					
		1000	100.0 ± 0.00 ^a^					
		1500	100.0 ± 0.00 ^a^					
	Nanoemulsion	Control	00.0 ± 00 ^e^	88.30 (77.06–100.43)	241.68 (198.93–317.40)	282.71 (235.99–359.86)	2.930 ± 0.301	6.266 (0.180)
		50	20.8 ± 2.33 ^d^					
		100	62.4 ± 3.25 ^c^					
		250	95.2 ± 3.88 ^b^					
		500	100.0 ± 0.00 ^a^					
		1000	100.0 ± 0.00 ^a^					
		1500	100.0 ± 0.00 ^a^					

a, b, c, d, e, f: There is no significant difference (*p* > 0.05) between any two means within the same column that have the same superscript letter (one-way ANOVA, Duncan’s MRT, *p* > 0.05).

**Table 3 plants-12-03682-t003:** Efficacy of larvicidal activity of patchouli oil and its nanoemulsion against *Culex pipiens* 24 h post-treatment.

Oil Name	Tested Materials	Conc. (ppm)	Mortality (%)	LC_50_ (Low–Up.)	LC_90_ (Low–Up.)	LC_95_ (Low–Up.)	Slope	Chi (Sig.)
Patchouli (*Pogostemon cablin*)	Oil	Control	00.0 ± 00 ^f^	276.29 (151.59–467.89)	954.25 (738.87–2569.01)	1355.99 (1131.15–4260.63)	2.381 ± 0.153	30.745 (0.000)
		50	8.0 ± 1.26 ^e^					
		100	18.4 ± 2.04 ^d^					
		250	34.4 ± 2.71 ^c^					
		500	60.0 ± 2.83 ^b^					
		1000	98.4 ± 1.60 ^a^					
		1500	100.0 ± 0.00 ^a^					
	Nanoemulsion	Control	00.0 ± 00 ^e^	93.05 (82.86–103.93)	221.18 (189.79–269.92)	321.52 (254.39–450.00)	3.408 ± 0.295	1.498 (0.827)
		50	16.8 ± 2.65 ^d^					
		100	56.8 ± 4.08 ^c^					
		250	91.2 ± 4.08 ^b^					
		500	100.0 ± 0.00 ^a^					
		1000	100.0 ± 0.00 ^a^					
		1500	100.0 ± 0.00 ^a^					

a, b, c, d, e, f: There is no significant difference (*p* > 0.05) between any two means within the same column that have the same superscript letter (one-way ANOVA, Duncan’s MRT, *p* > 0.05).

**Table 4 plants-12-03682-t004:** Efficacy of larvicidal activity of honeysuckle oil and its nanoemulsion against *Culex pipiens* 48 h post-treatment.

Oil Name	Tested Materials	Conc. (ppm)	Mortality (%)	LC_50_ (Low–Up.)	LC_90_ (Low–Up.)	LC_95_ (Low–Up.)	Slope	Chi (Sig.)
Honeysuckle (*Lonicera caprifolium*)	Oil	Control	00.0 ± 00 ^f^	130.63 (113.60–149.08)	431.51 (359.94–541.46)	605.47 (488.44–797.55)	2.469 ± 0.186	8.163 (0.086)
		50	16.8 ± 1.50 ^e^					
		100	41.6 ± 0.98 ^d^					
		250	66.4 ± 5.31 ^c^					
		500	98.4 ± 0.98 ^b^					
		1000	100.0 ± 0.00 ^a^					
		1500	100.0 ± 0.00 ^a^					
	Nanoemulsion	Control	00.0 ± 00 ^d^	56.22 (49.36–62.34)	110.87 (96.70–130.48)	134.40 (113.63–175.49)	4.346 ± 0.575	0.115 (0.998)
		50	41.60 ± 2.71 ^c^					
		100	85.6 ± 4.49 ^b^					
		250	100.0 ± 0.00 ^a^					
		500	100.0 ± 0.00 ^a^					
		1000	100.0 ± 0.00 ^a^					
		1500	100.0 ± 0.00 ^a^					

a, b, c, d, e, f: There is no significant difference (*p* > 0.05) between any two means within the same column that have the same superscript letter (one-way ANOVA, Duncan’s MRT, *p* > 0.05).

**Table 5 plants-12-03682-t005:** Efficacy of larvicidal activity of patchouli oil and its nanoemulsion against *Culex pipiens* 48 h post-treatment.

Oil Name	Tested Materials	Conc. (ppm)	Mortality (%)	LC_50_ (Low–Up.)	LC_90_ (Low–Up.)	LC_95_ (Low–Up.)	Slope	Chi (Sig.)
Patchouli (*Pogostemon cablin*)	Oil	Control	0.80 ± 0.80 ^f^	149.00 (105.78–201.73)	458.72 (352.75–757.44)	630.92 (484.37–129.25)	2.624 ± 0.181	12.040 (0.017)
		50	13.6 ± 2.04 ^e^					
		100	36.0 ± 1.79 ^d^					
		250	60.0 ± 3.79 ^c^					
		500	95.2 ± 3.88 ^b^					
		1000	100.0 ± 0.00 ^a^					
		1500	100.0 ± 0.00 ^a^					
	Nanoemulsion	Control	00.0 ± 00 ^d^	61.60 (54.30–68.44)	128.58 (111.28–158.77)	158.42 (132.88–206.85)	4.007 ± 0.481	1.369 (0.849)
		50	36.8 ± 4.63 ^c^					
		100	78.4 ± 4.12 ^b^					
		250	100.0 ± 0.00 ^a^					
		500	100.0 ± 0.00 ^a^					
		1000	100.0 ± 0.00 ^a^					
		1500	100.0 ± 0.00 ^a^					

a, b, c, d, e, f: There is no significant difference (*p* > 0.05) between any two means within the same column that have the same superscript letter (one-way ANOVA, Duncan’s MRT, *p* > 0.05).

**Table 6 plants-12-03682-t006:** Knockdown time and mortality rate of *Culex pipiens* mosquitoes exposed to 1% essential oils, 60 min.

Oil Name	1 h Knockdown %	LT_50_ (Low.–Up.)	LT_95_ (Low.–Up.)	RE (LT_50_)	Slope ± SE	Chi (Sig.)	24 h Mortality %
*Aloe vera*	22.22 ± 4.45 ^e^	168.94 (103.68–452.13)	1557.55 (54.18–14,099.81)	1.3	1.705 ± 0.312	0.693 (0.231)	42.22 ± 5.88 ^ef^
*Citrus aurantium*	51.11 ± 4.44 ^b^	66.37 (51.58–96.58)	619.03 (323.25–1782.13)	3.4	1.696 ± 0.211	3.740 (0.290)	60.00 ± 3.85 ^c^
*Cymbopogon schoenathus*	55.55 ± 2.22 ^a^	58.59 (45.42–85.29)	708.96 (351.95–2232.64)	3.8	1.519 ± 0.192	3.111 (0.374)	66.67 ± 7.7 ^b^
*Fragaria virginiana*	24.45 ± 2.22 ^e^	217.19 (114.01–849.20)	5703.56 (1260.65–15,676.66)	1.0	1.158 ± 0.223	0.592 (0.898)	37.78 ± 2.22 ^gh^
*Lactuca sativa*	17.78 ± 2.22 ^f^	225.56 (122–24-860.77)	2871.68 (780.99–54562.19)	1.0	1.488 ± 0.296	1.100 (0.296)	35.56 ± 4.44 ^h^
*Lavandula angustifolia*	37.78 ± 4.45 ^d^	108.30 (71.61–224.27)	2044.18 (704.05–14984.66)	2.1	1.289 ± 0.202	0.170 (0.982)	64.44 ± 5.88 ^b^
*Lonicera caprifolium*	51.11 ± 5.88 ^b^	69.32 (50.28–124.37)	1520.20 (568.23–8672.15)	3.3	1.238 ± 0.182	2.913 (0.971)	75.56 ± 4.44 ^a^
*Melissa officinalis*	33.33 ± 3.85 ^e^	193.27 (100.26–793.35)	8939.40 (1650.08–39785.89)	1.2	0.987 ± 0.195	1.348 (0.717)	48.89 ± 2.22 ^e^
*Narcissus tazetta*	53.33 ± 3.85 ^ab^	62.02 (47.31–93.30)	820.26 (388.48–2848.24)	3.6	1.466 ± 0.191	2.293 (0.513)	66.67 ± 7.70 ^b^
*Origanum majorana*	37.78 ± 2.22 ^d^	98.33 (66.92–190.18)	1732.27 (635.23–10,847.22)	2.3	1.320 ± 0.200	0.285 (0.962)	51.11 ± 5.88 ^e^
*Pelargonium graveolens*	51.11 ± 2.22 ^b^	64.34 (50.05–93.31)	628.10 (326.24–1821.33)	3.5	1.662 ± 0.207	1.425 (0.699)	55.55 ± 2.22 ^d^
*Plumeria rubra*	33.33 ± 3.85 ^d^	160.80 (89.07–540.98)	6671.87 (1405.54–19,163.43)	1.4	1.016 ± 0.191	0.531 (0.912)	40.00 ± 3.85 ^fg^
*Pogostemon cablin*	48.91 ± 5.88 ^b^	72.77 (52.68–130.66)	1527.80 (579.23–8681.75)	3.1	1.238 ± 0.182	2.913 (0.405)	73.33 ± 3.85 ^a^
*Punica granatum*	33.33 ± 3.85 ^d^	156.41 (87.16–518.08)	6638.84 (1401.15–18,896.49)	1.4	1.010 ± 0.190	0.096 (0.992)	44.45 ± 2.22 ^e^
*Ricinus communis*	33.33 ± 3.85 ^d^	93.46 (58.46–242.73)	2342.17 (637.70–38,422.49)	2.4	1.175 ± 0.221	0.216 (0.072)	40.00 ± 3.85 ^fg^
*Rosmarinus officinalis*	42.22 ± 8.01 ^c^	82.50 (60.27–136.28)	995.11 (445.22–3970.09)	2.7	1.521 ± 0.209	1.1971 (0.753)	64.44 ± 5.88 ^b^

Numbers in the same column followed by the same superscript letter are not significantly different (one-way ANOVA, Duncan’s MRT, *p* > 0.05); RE: relative efficacy.

**Table 7 plants-12-03682-t007:** Knockdown time and mortality rate of *Culex pipiens* mosquitoes exposed to 5% essential oils, 60 min.

Oil Name	1 h Knockdown %	LT_50_ (Low.–Up.)	LT_95_ (Low.–Up.)	RE (LT_50_)	Slope ± SE	Chi (Sig.)	24 h Mortality %
*Aloe vera*	37.78 ± 4.45 ^h^	122.78 (74.34–321.71)	4348.13 (1100.04–71,388.02)	1.0	1.061 ± 0.188	0.218 (0.974)	71.11 ± 4.44 ^g^
*Citrus aurantium*	91.11 ± 5.88 ^a^	26.56 (23.07–31.81)	158.05 (98.16–375.08)	4.6	2.123 ± 0.192	26.718 (0.000)	93.33 ± 6.67 ^c^
*Cymbopogon schoenathus*	91.11 ± 4.44 ^a^	24.36 (21.68–27.96)	160.48 (89.07–540.98)	5.0	2.009 ± 0.184	23.150 (0.000)	86.67 ± 3.85 ^d^
*Fragaria virginiana*	53.33 ± 3.85 ^g^	60.51 (45.50–93.69)	998.78 (439.54–4045.25)	2.0	1.350 ± 0.183	2.662 (0.446)	75.56 ± 5.88 ^f^
*Lactuca sativa*	55.55 ± 2.22 ^g^	57.84 (45.60–81.58)	584.43 (309.33–1623.07)	2.1	1.637 ± 0.199	0.730 (0.865)	73.33 ± 3.85 ^fg^
*Lavandula angustifolia*	75.56 ± 5.88 ^d^	30.07 (25.43–36.73)	276.07 (173.53–550.99)	4.1	1.708 ± 0.178	4.465 (0.215)	95.55 ± 2.22 ^bc^
*Lonicera caprifolium*	84.45 ± 2.22 ^b^	22.19 (19.68–26.96)	183.54 (124.87–319.54)	5.5	1.819 ± 0.177	6.765 (0.079)	100.00 ± 00 ^a^
*Melissa officinalis*	55.55 ± 2.22 ^g^	47.01 (36.45–68.13)	841.66 (384.26–3148.23)	2.6	1.312 ± 0.174	0.698 (0.873)	86.67 ± 3.85 ^d^
*Narcissus tazetta*	93.33 ± 3.85 ^a^	21.14 (17.66–24.98)	130.17 (67.16–355.08)	5.8	2.072 ± 0.184	17.977 (0.000)	97.78 ± 2.22 ^ab^
*Origanum majorana*	71.11 ± 4.44 ^e^	36.36 (25.55–82.14)	330.99 (266.18–4828.75)	3.4	1.714 ± 0.184	8.002 (0.046)	88.89 ± 4.44 ^d^
*Pelargonium graveolens*	80.00 ± 6.67 ^c^	28.93 (24.88–32.66)	270.04 (162.24–880.77)	4.2	1.695 ± 0.177	11.292 (0.010)	93.33 ± 3.85 ^c^
*Plumeria rubra*	55.55 ± 2.22 ^g^	56.72 (42.65–87.82)	1063.70 (454.13–4601.90)	2.2	1.292 ± 0.178	2.499 (0.475)	80.00 ± 3.85 ^e^
*Pogostemon cablin*	82.22 ± 4.45 ^bc^	23.15 (20.07–27.81)	201.63 (134.14–364.77)	5.3	1.760 ± 0.175	5.055 (0.167)	100.00 ± 0 ^a^
*Punica granatum*	62.22 ± 2.22 ^f^	42.02 (33.40–57.84)	645.68 (320.35–2028.58)	2.9	1.386 ± 0.174	1.575 (0.665)	86.67 ± 3.85 ^d^
*Ricinus communis*	64.44 ± 4.44 ^f^	39.25 (32.52–50.10)	372.86 (220.69–830.34)	3.1	1.682 ± 0.185	0.915 (0.821)	82.22 ± 5.88 ^e^
*Rosmarinus officinalis*	75.56 ± 5.88 ^d^	37.93 (26.88–85.14)	355.52 (296.28–4868.11)	3.2	1.692 ± 0.184	18.104 (0.000)	95.56 ± 4.44 ^bc^

Numbers in the same column followed by the same superscript letter are not significantly different (one-way ANOVA, Duncan’s MRT, *p* > 0.05); RE: relative efficacy.

**Table 8 plants-12-03682-t008:** Lethal time and mortality rate of female *Culex pipiens* mosquitoes exposed to 5% honeysuckle and patchouli oils and their nanoemulsions.

Oil Name	Tested Materials	1 h Knockdown%	LT_50_ (Low.–Up.)	LT_95_ (Low.–Up.)	Slope ± SE	Chi (Sig.)	24 h Mortality %
Honeysuckle	Oil	84.45 ± 2.22	22.93 (19.70–26.98)	185.83 (127.55–317.05)	1.810 ± 0.170	6.778 (0.148)	100.00 ± 0.00
	Nano	100.00 ± 0.00	13.04 (11.08–15.14)	90.06 (65.45–142.53)	1.959 ± 0.195	6.624 (0.157)	100.00 ± 0.00
Patchouli	Oil	82.22 ± 4.45	23.93 (20.44–28.37)	211.42 (141.94–373.36)	1.738 ± 0.165	5.907 (0.206)	100.00 ± 0.00
	Nano	100.00 ± 0.00	14.34 (8.58–21.66)	76.43 (67.68–257.68)	2.263 ± 0.189	19.397 (0.000)	100.00 ± 0.00

**Table 9 plants-12-03682-t009:** Injected polyphenol standards (structure and concentration) and polyphenol contents and the concentration of honeysuckle and patchouli essential oils.

Polyphenol Contents	Standards		Honeysuckle Oil		Patchouli Oil	
	Conc. (µg/mL)	Area	Conc. (µg/g)	Area	Conc. (µg/g)	Area
Gallic acid	15	171.65	70.95	32.48	47.45	21.72
Chlorogenic acid	50	373.44	0.00	0.00	0.00	0.00
Catechin	75	291.29	60.96	9.47	0.00	0.00
Methyl gallate	15	239.43	12.45	7.95	0.00	0.00
Coffeic acid	18	241.80	0.00	0.00	0.00	0.00
Syringic acid	17.2	208.30	141.62	68.61	0.00	0.00
Pyro catechol	40	523.90	8.77	4.59	0.00	0.00
Rutin	61	445.91	0.00	0.00	0.00	0.00
Ellagic acid	120	327.76	0.00	0.00	0.00	0.00
p-Coumaric acid	20	710.86	0.00	0.00	10.77	15.31
Vanillin	12.9	338.87	8152.26	8566.02	6.32	6.64
Ferulic acid	20	324.86	0.00	0.00	0.00	0.00
Naringenin	30	259.83	10.89	3.77	0.00	0.00
Daidzein	35	491.37	3116.16	1749.93	18.66	10.48
Quercetin	40	310.98	67.97	21.14	170.25	52.94
Cinnamic acid	10	459.44	2447.91	4498.72	1.79	3.29
Apigenin	50	619.44	0.00	0.00	174.82	86.63
Kaempferol	60	507.81	0.00	0.00	0.00	0.00
Hesperetin	20	334.36	0.00	0.00	0.00	0.00

**Table 10 plants-12-03682-t010:** The major chemical constituents of *Lonicera caprifolium* essential oil.

No.	RT	Compound Name	Area (%)	R. I.	M. F.	Classification
1	2.02	Cyclobutane, 1,1-dimethyl-2-octyl	0.33	913	C_14_H_28_	Cycloalkane
2	7.36	D-Limonene	0.54	1030	C_10_H_16_	Monoterpene
3	9.59	Linalool	1.19	1099	C_10_H_18_O	Monoterpene
4	10.06	Phenethyl alcohol	0.62	914	C_8_H_10_O	Phenyl
5	11.58	Acetic acid, phenylmethyl ester	1.53	1164	C_9_H_10_O_2_	Phenol
6	12.75	Estragole	0.78	1196	C_10_H_12_O	Phenylpropene
7	13.35	1,3-Dioxolane, 4-ethyl-4-methyl-2-pentadecyl	0.40	1842	C_21_H_42_O_2_	Heneicosylic acid
8	13.74	Citronellol	1.36	1135	C_10_H_20_O	Monoterpene
9	14.42	Linalyl acetate	1.25	1257	C_12_H_20_O_2_	Monoterpene
10	14.52	Geraniol	1.17	998	C10H_18_O	Monoterpene
11	21.33	α- Isomethyl ionone	3.96	1480	C_14_H_22_O	Sesquiterpene
12	24.80	Diethyl phthalate	24.85	1594	C_12_H_14_O_4_	Phthalic acid
13	26.47	1-(4-Isopropylphenyl)-2-methylpropyl acetate	1.54	1578	C_15_H_22_O_2_	Flavonoid
14	26.69	β-Ionone, methyl	15.76	1489	C_14_H_22_O	Sesquiterpene
15	26.87	β-Ionone	3.06	1456	C_14_H_22_O	Sesquiterpene
16	30.50	5,5-Dimethyl-2-(7-hydroxy-n-heptyl)-2-n-hexyl-1,3-dioxane	2.26	1784	C_19_H_38_O_3_	Fatty acid
17	30.77	Oxacycloheptadec-8-en-2-one, (8Z)	5.44	1925	C_16_H_28_O_2_	Ketone
18	31.11	9,12-Octadecadienoic acid (Z, Z)	5.49	2133	C_18_H_32_O_2_	Methyl ester
19	31.28	i-Propyl 12-methyl-tridecanoate	0.64	1750	C_17_H_34_O_2_	Tridecanoic acid
20	31.50	7-Acetyl-6-ethyl-1,1,4,4-tetramethyltetralin	21.69	1851	C_18_H_26_O	Phenol
21	35.54	Ethylene brassylate	4.67	1989	C_15_H_26_O_4_	Ketone

**Table 11 plants-12-03682-t011:** The major chemical constituents of *Pogostemon cablin* essential oil.

No	RT	Compound Name	Area (%)	R. I.	M. F.	Classification
1	5.93	β-Pinene	0.18	937	C_10_H_16_	Monoterpene
2	18.35	α-Copaene	0.21	1419	C_15_H_24_	Sesquiterpene
3	18.57	β-Patchoulene	3.80	1419	C_15_H_24_	Sesquiterpene
4	18.82	β-Elemene	1.45	1419	C_15_H_24_	Sesquiterpene
5	19.50	Cycloseychellene	0.90	1419	C_15_H_24_	Sesquiterpene
6	19.68	β-Caryophyllene	3.44	1351	C_15_H_24_	Sesquiterpene
7	20.23	α-Guaiene	15.80	1419	C_15_H_24_	Sesquiterpene
8	20.53	Seychellene	9.34	1419	C_15_H_24_	Sesquiterpene
9	20.78	α-Humulene	0.42	1419	C_15_H_24_	Sesquiterpene
10	20.91	Levo-alpha-cedrene	6.73	1419	C_15_H_24_	Sesquiterpene
11	20.99	Valencene	1.70	1419	C_15_H_24_	Sesquiterpene
12	21.80	γ -Gurjunene	0.56	1419	C_15_H_24_	Sesquiterpene
13	22.05	Aciphyllene	3.31	1419	C_15_H_24_	Sesquiterpene
14	22.24	α-Bulnesene	16.88	1419	C_15_H_24_	Sesquiterpene
15	22.70	α-Selinene	0.40	1419	C_15_H_24_	Sesquiterpene
16	24.11	Norpatchoulenol	0.82	1480	C_14_H_22_O	Sesquiterpene
17	24.26	Diepicedrene-1-oxide	0.41	1496	C_15_H_24_O	Sesquiterpene
18	24.50	Caryophellene oxide	0.63	1496	C_15_H_24_O	Sesquiterpene
19	24.67	Spathulenol	0.57	1496	C_15_H_24_O	Sesquiterpene
20	25.57	Ledene oxide-(II)	0.65	1496	C_15_H_24_O	Sesquiterpene
21	25.86	Globulol	0.96	1583	C_15_H_26_O	Phenol
22	26.71	Pogostole	2.58	1583	C_15_H_26_O	Phenol
23	26.94	Patchouli alcohol	26.62	1660	C_15_H_26_O	Phenol
24	28.28	Pogostone	0.98	1787	C_12_H_16_O_4_	Ketone

**Table 12 plants-12-03682-t012:** Plant species screened (No. of oils = 16) for use for larvicidal activity.

No.	Oil Name	Plant Oils			
		Order	Family	English Name	Part Used
1	*Aloe vera* L.	Asparagales	Xanthorrhoeaceae	Mediterranean aloe	Leaf
2	*Citrus aurantium* L.	Sapindales	Rutaceae	Bitter orange	Flower
3	*Cymbopogon schoenanthus* L.	Poales	Poaceae	Camel grass	Leaf
4	*Fragaria virginiana* D.	Rosale	Rosaceae	Wild strawberry	Leaf/fruit
5	*Lactuca sativa* L.	Asterales	Asteraceae	Lettuce	Leaf
6	*Lavandula angustifolia* M	Lamiales	Lamiaceae	English lavender	Leaf
7	*Lonicera caprifolium* L.	Dipsacales	Caprifoliaceae	Honeysuckle	Flower
8	*Melissa officinalis* L.	Lamiales	Lamiaceae	Common balm	Leaf/flower
9	*Narcissus tazetta* L.	Asparagales	Amaryllidaceae	Cream narcissus	Flower
10	*Origanum majorana* L.	Lamiales	Lamiaceae	Sweet marjoram	Leaf
11	*Pelargonium graveolens*	Geraniales	Geraniaceae	Scented geranium	Leaf
12	*Plumeria rubra* L.	Gentianales	Apocynaceae	Frangipani	Flower
13	*Pogostemon cablin* B.	Lamiales	Lamiaceae	Patchouli	Leaf
14	*Punica granatum* L.	Myrtales	Lythraceae	Pomegranate	Flower
15	*Ricinus communis* L.	Malpighiales	Euphorbiaceae	Castor bean	Seed
16	*Rosmarinus officinalis* L.	Lamiales	Lamiaceae	Rosemary	Flower

Plant oils purchased from the Nefertari Company for extracting natural essential oils and body care products.

## Data Availability

Not applicable.

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
