# Peer review of "Sustainable Pest Management Using Novel Nanoemulsions of Honeysuckle and Patchouli Essential Oils against the West Nile Virus Vector, Culex pipiens, under Laboratory and Field Conditions"

_plants, 2023, doi:10.3390/plants12213682_

Round 1
Reviewer 1 Report
The presented data provides good information on EOs from plants as an alternative control strategy for managing mosquito species. The results appear reasonable and are generally well-written. Please address comments or questions to be considered for publication. See the attached file for the comments.

Author Response
Thanks for Reviewer 1 for careful revising of the manuscript
Thanks to the editor and reviewers for though revising our work
- General corrections
- The abstract has been shortened as appropriate
- More recent information was added to the text sections.
- All manuscript sections were improved as recommended
- Corrections were highlighted in yellow
- A statement of the supplementary files was mentioned at the end of the manuscript before the references

Reviewer 2 Report
This is an interesting study focusing on the larvicidal and other properties of honeysuckle and patchouli essential oils. The research is well designed. I think some of the tables can be moved to supplemental figures section while other tables can be combined into one. Some information in the Discussion section is repeated unnecessarily, can be removed. Multiple typos and grammatical errors. I am attaching the file below with my comments.

I do not have nay major concerns with the manuscript as such. The research is well designed and can be of interest to people. Minor changes need to be made in English writing throughout the manuscript. Some tables should be moved to the supplemental section while those in the initial part of the results section can be combined. Also they can be put in graphs which makes it more interesting in stead of tables.
Author Response
Thanks for Reviewer 2 for careful revising of the manuscript
Thanks to the editor and reviewers for though revising our work
- General corrections
- The abstract has been shortened as appropriate
- More recent information was added to the text sections.
- All manuscript sections were improved as recommended
- Corrections were highlighted in yellow
- A statement of the supplementary files was mentioned at the end of the manuscript before the references
